# Dissection of two routes to naïve pluripotency using different kinase inhibitors

Ana Martinez-Val [1,5], Cian J. Lynch[2], Isabel Calvo [2], Pilar Ximénez-Embún [1,3], Fernando Garcia[1,3], Eduardo Zarzuela[1,3], Manuel Serrano [2,4] & Javier Munoz [1,3 ✉]

Embryonic stem cells (ESCs) can be maintained in the naïve state through inhibition of Mek1/2 and Gsk3 (2i). A relevant effect of 2i is the inhibition of Cdk8/19, which are negative regulators of the Mediator complex, responsible for the activity of enhancers. Inhibition of Cdk8/19 (Cdk8/19i) stimulates enhancers and, similar to 2i, stabilizes ESCs in the naïve state. Here, we use mass spectrometry to describe the molecular events (phosphoproteome, proteome, and metabolome) triggered by 2i and Cdk8/19i on ESCs. Our data reveal widespread commonalities between these two treatments, suggesting overlapping processes. We find that post-transcriptional de-repression by both 2i and Cdk8/19i might support the mitochondrial capacity of naive cells. However, proteome reprogramming in each treatment is achieved by different mechanisms. Cdk8/19i acts directly on the transcriptional machinery, activating key identity genes to promote the naïve program. In contrast, 2i stabilizes the naïve circuitry through, in part, de-phosphorylation of downstream transcriptional effectors.

[1] Proteomics Unit, Spanish National Cancer Research Centre (CNIO), Madrid, Spain. [2] Institute for Research in Biomedicine (IRB Barcelona), Barcelona Institute of Science and Technology (BIST), Barcelona, Spain. [3] ISCIII-ProteoRed, Madrid, Spain. [4] Catalan Institution for Research and Advanced Studies (ICREA), Barcelona, Spain. [5] Present address: The Novo Nordisk Foundation Center for Protein Research, University of Copenhagen, Copenhagen, Denmark. ✉email: jmunozpe@cnio.es

A spectrum of transitory pluripotent states occurs during early development, and these can be captured in vitro through stimulation and blockade of specific pathways. Leukemia inhibitory factor (LIF) together with inhibitors of Mek1/2 and Gsk3 (2i)[1] stabilize mouse embryonic stem cells (mESCs) into the so-called naïve state, which closely resembles the pre-implantation blastocyst[2]. Recently, we showed that naïve pluripotency can also be stabilized by inducing a global upregulation of key identity genes under the control of super-enhancers[3]. This is achieved by inhibiting Cdk8/19 (Cdk8/19i) and, thereby, relieving their negative effect on the Mediator complex at enhancers. This suggests the existence of common regulatory processes involved in the acquisition of naïve pluripotency conferred by 2i-treatment or Cdk8/19i-treatment. Understanding the pathways that regulate pluripotency is important, especially for the derivation of naïve human ESCs[4].

Compared to the more developmentally advanced states of the post-implantation embryo (i.e., epiblast stem cells (EpiSCs), known as primed state cells; or serum ESCs, which represent a metastable state that fluctuates between primed and naïve pluripotency), naïve ESCs express a set of transcription factors that reinforce the pluripotency core factors while repressing lineage specification genes[5]. Furthermore, while primed mEpiSCs and hESCs are mainly glycolytic, naïve mESCs use oxidative phosphorylation[6]. Another key feature of naïve cells is their hypomethylated DNA[7], although this has deleterious implications upon long-term passages[8]. Importantly, a profound connection between the metabolism of these cells and their epigenetic status emerges as several metabolites have been found to determine histone and DNA methylation levels[9].

Despite the above-mentioned transcriptomic and epigenetic insights, our understanding of pluripotency demands direct analysis of protein-driven mechanisms, yet crucially, this has been less explored[10–12]. Post-transcriptional regulation makes mRNA levels an imperfect predictor of protein abundance in mESCs[13]. Protein function is also rapidly altered by post-translational modifications (PTMs), and methods used to stabilize naïve cells target specific kinases, but how signaling is connected to transcriptional control remains unclear. In this work, we use mass spectrometry (MS) to dissect the establishment of naïve pluripotency by two methods: blunting extracellular signaling (2i) and stimulating enhancer transcription (Cdk8/19i). Using an integrative strategy, we analyze three interconnected regulatory layers. First, we profile the early phosphorylation cascades that are triggered in response to these inhibitors (0–6 h) to identify downstream effectors. Next, we monitor how protein levels are tuned until naïve pluripotency is stabilized (0–14 days). In addition, since the availability of metabolites can determine their epigenetic status, we also probe the metabolome of cells adapted to 2i or Cdk8/19i conditions. Our analyses reveal missing links in the molecular roadmap from primed to naïve pluripotency, and provide insights into the regulatory principles of cell identity.

## Results

### Stabilization of naïve pluripotency by 2i and Cdk8/19i results in similar proteome changes.
To study proteome remodeling during the stabilization of mESCs in the naïve state, we cultured a panel of four mESC lines in standard serum/LIF (S/L) media. Cells were then supplemented with either 2i or Cdk8/19i, collected after 1, 2, 4, 7, 10, or 14 days of treatment, and subjected to proteomic analysis (Fig. 1a). We chose to keep serum in all our experimental conditions to better isolate and therefore study the events regulated exclusively by the 2i and Cdk8/19i (Supplementary Note 1). Nevertheless, we have previously shown that the

effects of Cdk8/19i in terms of morphology, homogenous expression of Nanog-GFP and transcriptomic changes are highly similar between serum-containing and serum-free media[3]. Using isobaric labeling, a total of 4408 proteins were quantified in all 64 samples (Supplementary Fig. 1A), enabling us to accurately monitor their temporal kinetics. PCA grouped all S/L samples together, indicating the high reproducibility of our data (Fig. 1b and Supplementary Fig. 1B). Cells treated for 1 day with either 2i or Cdk8/19i clustered separately from S/L implying that, as early as 24 h, their proteomes are already distinct from S/L, in agreement with transcriptomics[14]. Of note, PCA trajectories suggested that 2i and Cdk8/19i induced, to a certain extent, similar changes. Upon 24 h of 2i and Cdk8/19i, 717 and 444 proteins increased their expression compared to S/L respectively. Importantly, 219 of them were common to both treatments ($p.val = 8.8E-68$) (Fig. 1c and Supplementary Data 1). Conversely, proteins showing opposite changes in 2i and Cdk8/19i were under-represented ($p.val = 2.7E-20$) (Supplementary Fig. 1C). The number of changes increased over time until day 7, when 47% of the proteome in 2i and 37% in Cdk8/19i showed differential regulation. Throughout all time-points, the regulated proteins overlapped significantly between 2i and Cdk8/19i, with minor inverse changes (Fig. 1c and Supplementary Fig. 1C). Next, we mapped transcriptomic signatures of 2i-treated cells in our temporal-resolved proteome data and found that mRNA changes showed overall concomitant protein alterations after 48 h of 2i treatment, which were also recapitulated, to some degree, in Cdk8/19i (Supplementary Fig. 1D). Likewise, post-implantation genes correlated with repressed proteins in both 2i and Cdk8/19i, whereas pre-implantation genes (E3.5 and E4.5) showed positive trends (Supplementary Fig. 1D). These results indicate that stabilization of the naive state by these two chemical methods involves the regulation of a large repertoire of proteins, some of which are common to both strategies but also numerous proteins that are regulated in a specific manner. In addition, both 2i and Cdk8/19i recapitulate in vitro and in vivo transcriptomic signatures.

Several proteins involved in pluripotency (Tbx3, Klf4, and Esrrb) and the naïve circuitry (Tfcp2l1) showed similar up-regulation in 2i and Cdk8/19i. Lin28a, Dppa2, Dnmt3a, and Eras, that are required in the primed state, were downregulated upon inhibition of both pathways (Fig. 1d and Supplementary Fig. 1E). We found that most of the proteome exhibited positive correlation between 2i and Cdk8/19i, indicating that a large fraction of the proteome undergoes similar temporal regulation (Fig. 1e). We used unsupervised clustering and found 15 co-regulated protein modules exhibiting various kinetics (Fig. 1f). The two clusters containing more proteins, #7 (388 proteins) ($r = 0.91$; $p.val = 0.039$) and #15 (401 proteins) ($r = 0.97$; $p.val = 0.0002$), displayed remarkable similar patterns in 2i and Cdk8/19i. Nevertheless, we also found clusters exhibiting different dynamics between the treatments (#9, #12, #6) as well as clusters of proteins that were only co-regulated in 2i (#3, #1, #2) (Fig. 1f). Among the clusters containing upregulated proteins in both treatments (#7), we found enrichment of proteins involved in aerobic respiration and beta-oxidation (Fig. 1g). In contrast, terms related to translation appeared enriched among proteins with downregulated trends in both inhibitors (#15) (Fig. 1g). In addition, proteins related to adherent junctions (#9) or RNA processing (#3) were found in proteins displaying treatment-specific kinetics. Together, our results indicate that acquisition of naïve pluripotency by 2i and Cdk8/19i invokes multiple biological processes (Supplementary Data 2). Some of these processes displayed similar temporal kinetics, underscoring the presence of shared mechanisms. However, our data also showed the existence of proteins regulated exclusively by each treatment, indicating unique regulatory mechanisms too.

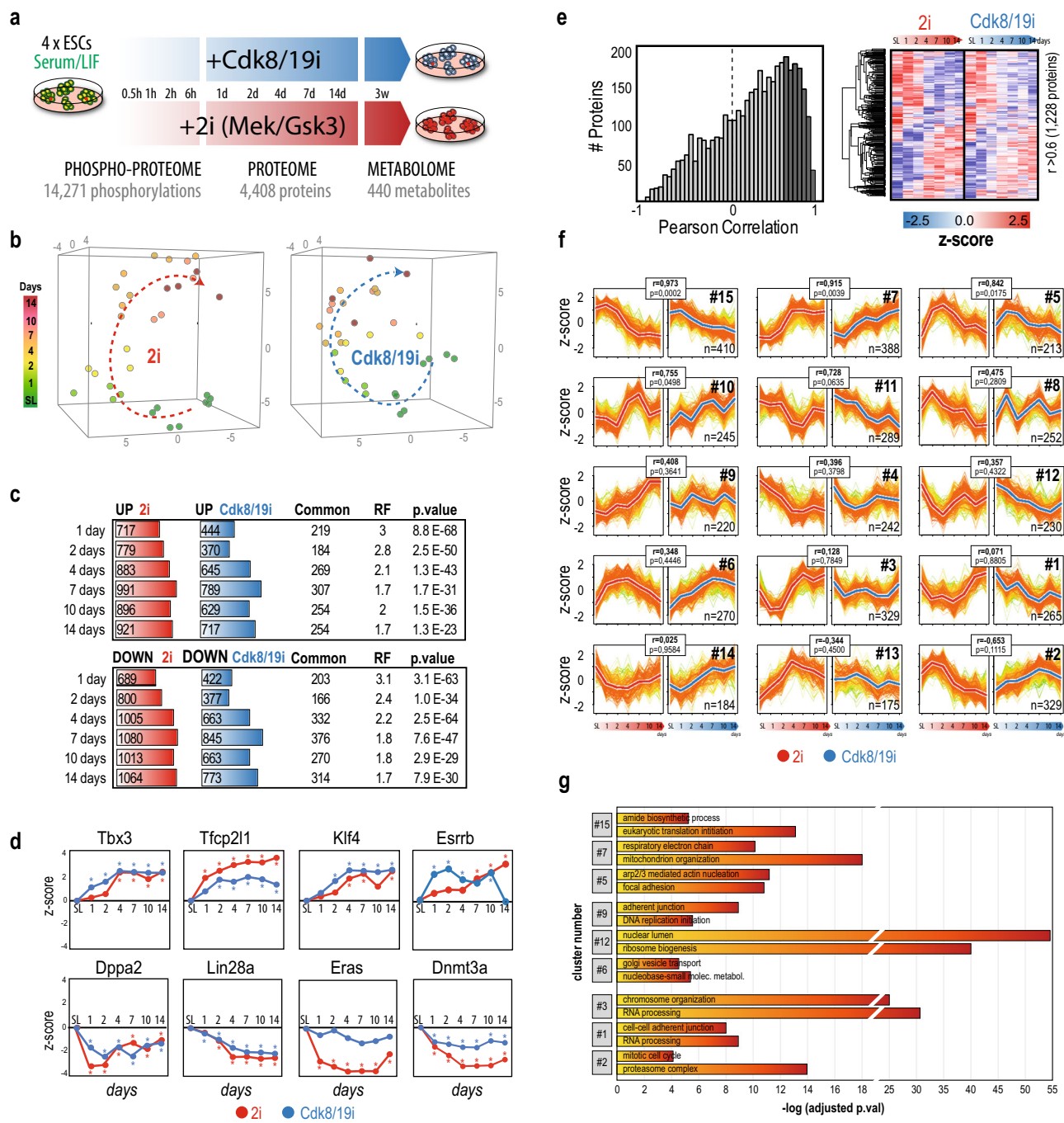

**Fig. 1 Time-resolved proteome profiling of mESCs transitioning to the naïve state using 2i and Cdk8/19i. a** Four different mESCs were grown in Serum/LIF and transferred to Cdk8/19i or 2i supplemented media. Samples were recovered at different time points to measure the changes in the phosphoproteome, proteome and metabolome. **b** Principal Component Analysis of the proteomes of mESCs treated with 2i (red) or Cdk8/19i (blue) at different time points. **c** Differentially regulated proteins using a moderated *t*-test (limma two-sided, FDR < 5%) at the indicated time points. The number of common proteins between 2i and Cdk8/19i is shown, together with the corresponding *p*.val calculated by a hypergeometric test (RF indicates "Representation factor"). **d** Examples of changes in protein abundance for proteins related to pluripotency in response to 2i or Cdk8/19i (*N* = 4 independent cell lines). Statistically significant changes using a moderated *t*-test (limma two-sided, FDR < 5%) relative to reference S/L are indicated with an asterisk (*p*-value < 0.05). **e** Histogram of the Pearson correlation values (Pearson's correlation test, two-tailed) between the temporal proteins trends of 2i and Cdk8/19i. The 1228 proteins with *r* > 0.6 (dark gray) are displayed in the heatmap. **f** Gap statistics and K-means classified the regulated proteome into 15 different clusters. The Pearson correlation and the corresponding *p*.val (Pearson's correlation test, two-tailed) between the centroids (thick lines) of 2i (red) and Cdk8/19i (blue) are shown. Color gradient in the line plots indicate proximity to the centroid of the cluster. **g** Examples of GO-enriched terms in examples of clusters showing similar (#15, #7, #5) and different regulation (#9, #12, #6) between 2i and Cdk8/19i, as well as in clusters showing 2i-specific changes (#3, #1, #2). All the results from GO-enrichment analyses can be found in Supplementary Data 2. Source data for Fig. 1b, e are provided as a Source Data file.

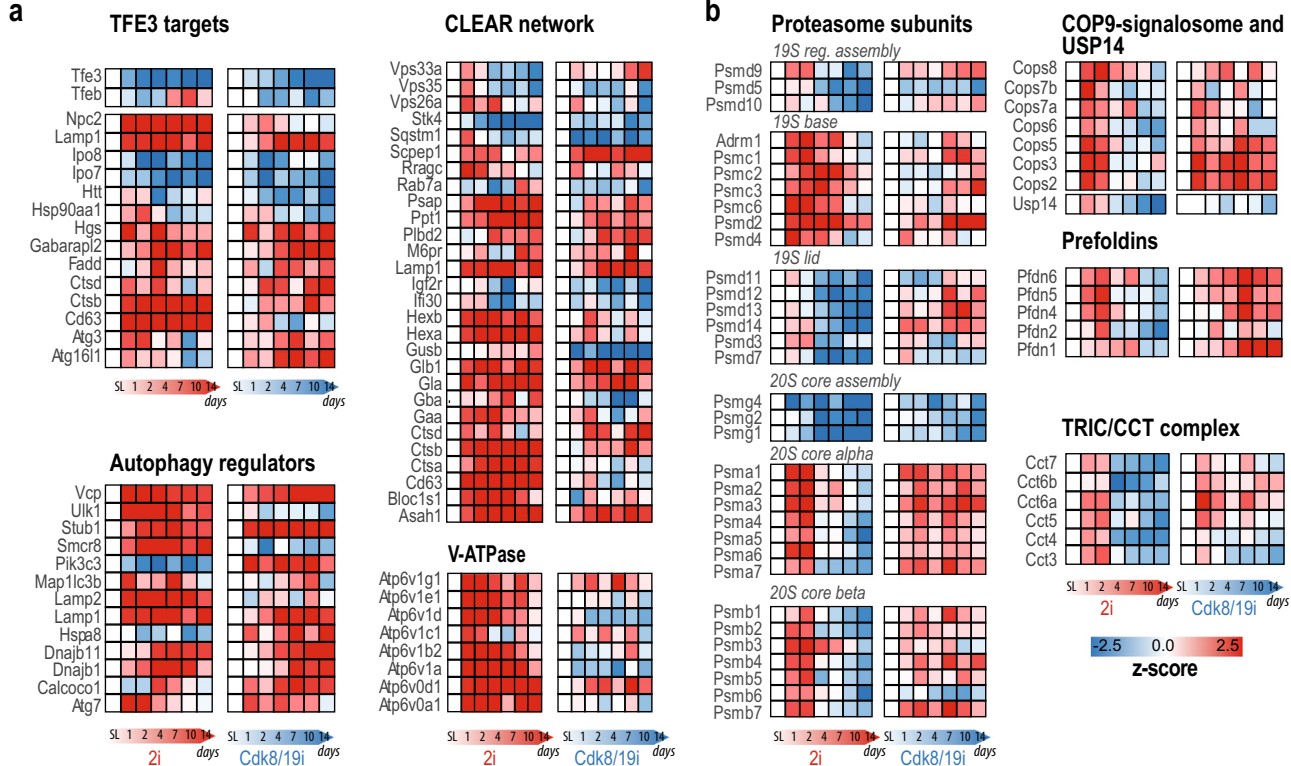

**Fig. 2 Differential regulation of the autophagy and proteolysis machinery.** Heatmaps showing the 2i-time and Cdk8/19i-time dependent protein changes for several components involved in autophagy (**a**) as well as protein complexes involved in proteostasis (**b**). Source data are provided as a Source Data file.

**Upregulation of proteostasis machineries during the transition to naïve pluripotency**. The most upregulated protein after 14 days of 2i, Gabarapl2, was also the highest change in Cdk8/19i (Fig. 2a and Supplementary Data 1). Gabarapl2 is involved in autophagy, a process that has multiple links to pre-implantation[15], diapause[16], and pluripotency[17]. The master transcription factor of autophagy Tfe3 rapidly translocates to the nucleus in response to 2i[18], and we have shown that Cdk8/19i recapitulates this event[3]. Tfe3, together with Tfeb and Mitf, belong to the MiTF/Tfe family that bind the CLEAR elements present in the promoters of lysosomal biogenesis genes[19,20]. We then examined the protein dynamics of the MiTF/Tfe target genes in our data. Several lysosomal hydrolases such as Ctsb, Glb1, Asah1, Ppt1, and Scpep1 increased in both 2i and Cdk8/19i cultures compared to reference S/L (Fig. 2a). However, the expression of CLEAR-box proteins was more significant in 2i. Likewise, 2i, but not Cdk8/19i, increased the abundance of all the subunits of the V-ATPase involved in the acidification of the lysosomal lumen (Fig. 2a). We found similar trends in other 2i published data sets (Supplementary Fig. 2C). To understand these differential changes in 2i and Cdk8/19i, we checked the levels of the MiTF/Tfe factors. Tfe3 protein levels decreased in response to both drugs, probably reflecting the selective degradation of its cytoplasmic form[21] (Fig. 2a). Tfeb on the other hand did not show significant changes (Fig. 2a). However, Mitf mRNA expression was remarkably different: 2i increased its levels 6-fold whereas it remained unaltered in Cdk8/19i (Supplementary Fig. 2A). In agreement with our data, Mitf is reported to regulate the expression of the V-ATPase pump[22] and it increases upon ERK inhibition[23]. Further, we found increased levels of LC3-II by western blot (Supplementary Fig. 2B), which could reflect either enhanced autophagosome synthesis or decreased autophagosome degradation. We noticed however that this increase was more evident in 2i than in Cdk8/19i, which might explain some of the proteomic differences found in autophagy components between both treatments. Therefore, our results show that naïve cells upregulate several autophagic components, likely through control of members of the MiTF/Tfe family.

Unlike degradation of autophagy, the ubiquitin-proteasome system (UPS) is the major degradation pathway for short-lived proteins. Our data revealed a co-regulation of proteasome subunits only in 2i (Cluster #2, $q$.val = 1.2E−14) (Supplementary Data 2). Within 48 h, a transitory increase in the 20S and 19S particles was found with the exception of the lid subunits that remained unaltered (Fig. 2b). Interestingly, the signalosome COP9 complex, that has been proposed to replace the lid in the formation of specialized proteasomes[24], exhibited a similar transient kinetic (Fig. 2b). After 4 days, the proteasome and COP9, including several assembly factors, decreased, suggesting a rapid induction of these complexes followed by their disassembly/ degradation (Fig. 2b). Besides autophagy and the UPS, chaperones also control proteome homeostasis. We found that the chaperonin TRiC/CCT and Prefoldin complexes were tightly co-regulated with the proteasome (Fig. 2b). Using a recent proteome-wide co-regulation analysis[25], we confirmed that all these macromolecular machineries are frequently co-regulated (Supplementary Fig. 2D, E). Interestingly, cluster#3 in 2i, which mainly contained nuclear proteins ($q$.val = 1.18E−65), displayed an opposite kinetic to the proteostasis machinery ($r = -0.94$, $p$.val = 1.4E−3) (Fig. 1f). Rapid degradation of nuclear proteins has been postulated as a mechanism for gene expression control, and all these complexes have been found in active genomic regions[26,27]. Nonetheless, whether the concerted regulation of these macromolecular complexes is linked to the degradation of nuclear proteins will require further experimentation.

**Prevalent post-transcriptional regulation in naïve pluripotency.** Another wave of proteome remodeling occurred around 4–7 days and involved the upregulation of a large fraction of proteins that

remained increased afterwards (cluster#7) (Fig. 1f). Importantly, these changes were highly synchronous in 2i and Cdk8/19i (p.val = 0.039) indicating a central role for these proteins in naïve pluripotency. GO analyses revealed a massive representation of mitochondrial proteins (q.val = 4.9E−72) (Supplementary Data 2) (Fig. 1g). We found a clear increase in all major electron transport chain (ETC) complexes with the exception of complex-I that was primarily upregulated in 2i (Fig. 3a). Complex I is the entry point

of the redox process and generates up to 40% of the ATP in the cell. Therefore, we questioned whether this difference in the stoichiometry of ETC complexes had any impact on ATP production. In agreement with our proteomic results, we found that 2i cells were more sensitive to rotenone, a potent complex I inhibitor, than S/L cells, indicating that they rely on Complex I as the main source for ATP (Fig. 3b). On the other hand, Cdk8/19i treated cells were resistant to this drug, implying that these cells are less

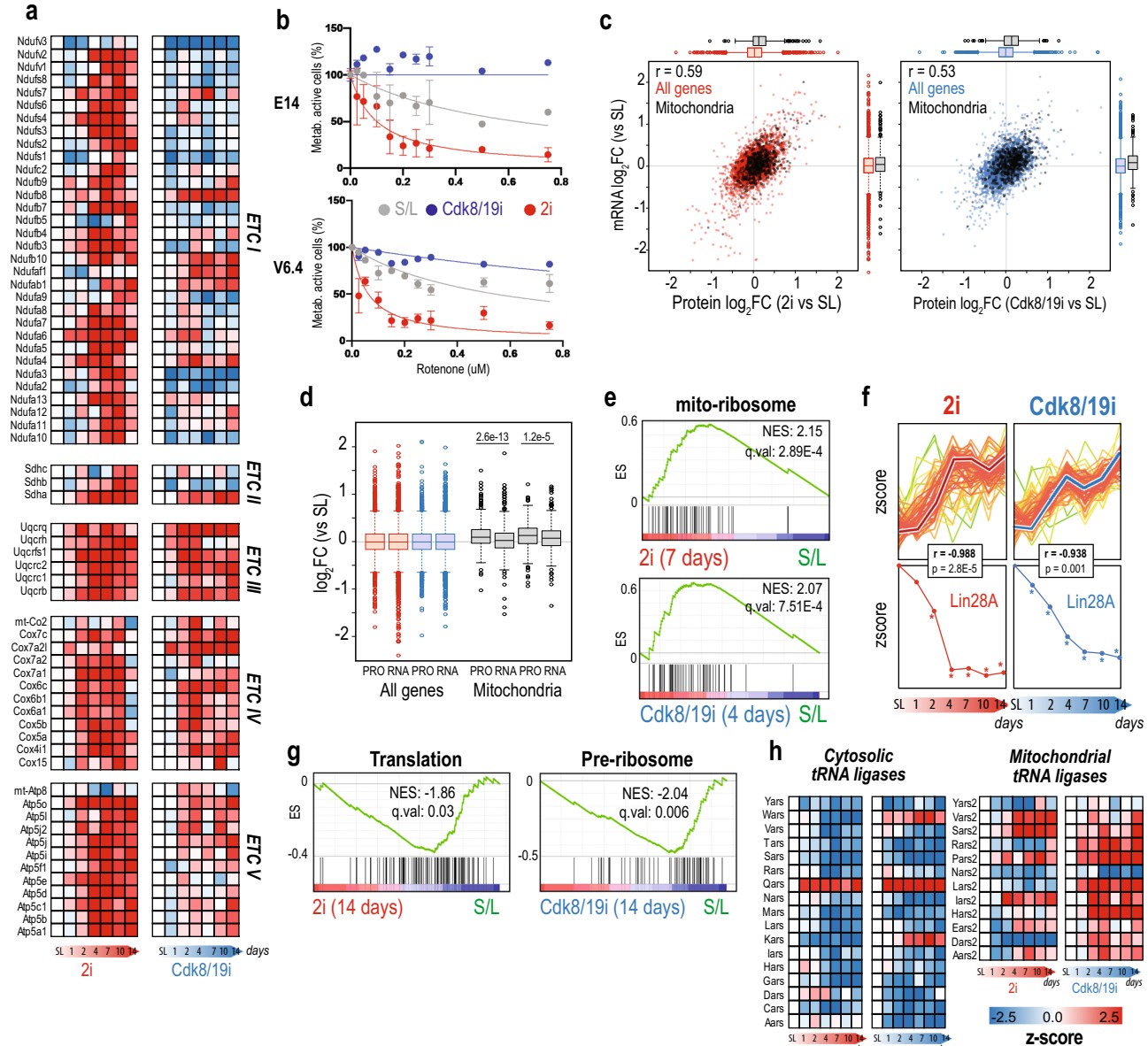

**Fig. 3 Degradation of LIN28a in 2i and Cdk8/19i releases target mRNAs and enhances translation of mitochondrial components. a** Protein levels across time for the electron transport chain subunits. **b** Rotenone induced dose-dependent cytotoxicity on S/L, 2i and Cdk8/19i mESCs in two different cell lines (E14 and V6.4). Data is expressed as percentage of ATP in non-treated wells, and it is shown as mean (n = 3 technical replicates) ± st dev. **c** Correlation of transcriptomic and proteomic changes for 2i (red) and Cdk8/19i (blue) relative to S/L. In black, mitochondrial proteins (UniProt). Boxplots represent the distribution of transcriptomic or proteomic log2 ratios of 2i (red) or Cdk8/19i (blue) vs. SL for total proteins (n = 4093 proteins) and mitochondrial proteins (black, n = 515 proteins). **d** Boxplots for protein (n = 4 independent cell lines) and mRNA (n = 3 biologically independent samples) ratios (2i or Cdk8/19i vs. S/L) for all genes (light, n = 4093 proteins) and for proteins localized in the mitochondria (black, n = 515 proteins). Statistical significance was calculated using a paired two-sample t-test for means, two-sided. Boxplots show medians and limits indicate the 25th and 75th percentiles, whiskers extend 1.5 times the interquartile range from the 25th and 75th percentiles, outliers are represented by dots. **e** Upregulation of mitoribosome subunits at the indicated time points and treatments by GSEA. **f** Mitochondrial components showing post-transcriptional regulation (above) are synchronized with the downregulation of Lin28A protein (below). n = 4 independent cell lines (**g**) GSEA for gene sets related to translation at 14 days of treatment with 2i or Cdk8/19i. **h** Protein changes of the cytosolic and mitochondrial tRNA ligases. Source data for Figs. 3a, b, d, h are provided as a Source Data file.

dependent on Complex I and that they may have alternative sources of ATP (Fig. 3b). Interestingly, we noticed that two important Complex I assembly factors (Ndufaf1 and Ndufaf7) were differentially regulated in Cdk8/19i (Supplementary Data 1), which might cause defects in its assembly process and might explain the results observed in our data for this inhibitor.

It has been reported that naïve cells possess immature mitochondria and perinuclear localization[28]. Using transmission electron microscopy, we confirmed such distinct features but did not appreciate differences in the size nor the morphology of mitochondria in 2i, Cdk8/19i cells compared to reference S/L (Supplementary Fig. 3A). Furthermore, we used MitoTracker staining and found similar number of mitochondria in S/L, 2i and Cdk8/19i conditions (Supplementary Fig. 3B, C), indicating that the upregulation of mitochondrial proteome observed in our data responded mainly to actual changes in the protein content rather than an increase of the mitochondrial biomass. Next, we investigated the origin of the augmented mitochondrial proteome in our data. Besides transcription regulation by PGC-1α[29], post-transcriptional regulation plays a role in tuning the levels of mitochondrial proteins[30,31]. To test the predominant regulatory mode in naïve pluripotency, we compared our proteome data with matched RNA-seq[3]. This analysis showed an overall good correlation between the transcriptomes and proteomes (Fig. 3c). However, mitochondrial protein levels were significantly higher than their mRNAs in both approaches ($p$.val-2i = 3E−13; $p$.val-Cdk8/19i = 1E−05) (Fig. 3c, d). We did not find such discrepancy for other organelles (Supplementary Fig. 3D). We confirmed these findings using other RNA-seq and proteome data sets of 2i mESCs (Supplementary Fig. 3E and Supplementary Note 2). Notably, nearly all the subunits of the mitoribosome peaked expression around 4–7 days, coincident with the maximum increase of mitochondrial components (Fig. 3e). These results indicate that the increase in mitochondrial proteins of 2i and Cdk8/19i cells as they transition into the naïve state is controlled, in part, post-transcriptionally.

mTOR has been linked to mitochondrial biogenesis by promoting translation via inhibition of 4E-BPs[30]. Among the targets of mTOR-enhanced translation is Tfam. However, TFAM mRNA and protein levels remained unaltered in our data (Supplementary Fig. 3F). On the other hand, Lin28 regulates metabolism and transition to primed pluripotency by repressing translation of mRNAs involved in OXPHOS[31]. Indeed, we found several proteins regulated in a post-transcriptional manner that showed opposite kinetics to Lin28 in response to both treatments. Consistent with this, several mitochondrial proteins mirrored the downregulation of Lin28 in a significant manner (Fig. 3f and Supplementary Fig. 3G). In addition to OXPHOS, Lin28 binds other mRNAs belonging to splicing factors[32] and endoplasmic reticulum[33]. We found 385 (2i) and 387 (Cdk8/19i) proteins (93 in common, $p$.val = 1.0E−19) which increased levels were not accompanied by changes in their cognate mRNAs (Supplementary Data 3). Indeed, these genes were enriched in RNA splicing ($q$.val-2i = 2.2E−17, $q$.val-Cdk8/19i = 4.4E−4) and the endomembrane system ($q$.val-2i = 1.4E−12, $q$.val-Cdk8/19i = 0.006) (Supplementary Data 3). Therefore, our results seem to agree with previous studies reporting post-transcriptional regulation in mESCs through Lin28-dependent degradation[31–33].

Despite above the evidence of enhanced translation, we noticed numerous proteins involved in protein synthesis, ribosome biogenesis, and tRNA processing that were downregulated in 2i and Cdk8/19i (Fig. 3g). These findings may suggest that naïve ESCs possess lower global translation rates than conventional S/L. In fact Myc depletion and mTOR inhibition induce a dormant state marked by a shutdown of protein synthesis[16,34]. This global reduction may balance the need of synthesis of mitochondrial proteins (see "Discussion" section). In support of this hypothesis, we found that, unlike cytosolic tRNA synthetases, their mitochondrial counterparts were predominantly upregulated (Fig. 3h).

**Culture conditions define specific metabolic signatures in mESCs.** Our proteomic data suggested changes in mitochondrial metabolism, but also in many other routes (Supplementary Data 2). Using LC-MS/MS, we profiled the metabolomes of four mESC lines grown for 2 weeks in S/L, 2i, or Cdk8/19i. Each cell line was analyzed in quadruplicates. This set up led to the quantification of 440 metabolites across all 48 samples. PCA analysis separated samples based on culture conditions (Fig. 4a). Differential analysis showed 184 metabolites downregulated in 2i and 115 in Cdk8/19i (Supplementary Fig. 4A and Supplementary Data 4). Among them, 65 were found in both inhibitors ($p$.val = 2E−4). In contrast, the number of upregulated compounds was lower: 50 in 2i and 66 in Cdk8/19i, with few metabolites in common. These results suggest that cells in 2i and Cdk8/19i diverge partially in their metabolic state, which also differs from reference S/L mESCs.

For instance, our data pointed out to differences in the metabolism of amino acids and nucleotides between these two chemical approaches. Unlike other proliferative cells, ESCs in 2i grow in the absence of glutamine[35]. We found that glutamine synthase (Glul) was actually repressed in 2i whereas it was overexpressed in Cdk8/19i (Fig. 4b). Accordingly, glutamine and glutamate levels differed in 2i and Cdk8/19i. Glutamine metabolism is closely linked to asparagine and, as expected, we observed differences in asparagine and aspartate levels as well as in the expression of the asparagine synthetase (Asns) (Fig. 4b). Of note, Asns is a key target of Atf4, a transcription factor that is activated in response to amino acid imbalance[36]. Most of the Atf4 targets increased in 2i including proline enzymes[37] (Supplementary Fig. 4B). In agreement, Atf4 and two critical effectors, Ddit3 and Eif2ak4 were only up-regulated in 2i (Supplementary Fig. 4C) indicating that amino acid metabolism differs in this aspect between 2i and Cdk8/19i. Furthermore, glutamine enables rapid proliferation because of its role in multiple biosynthetic pathways including nucleic acids. Our data revealed that mono-phosphorylated nucleotides decreased in 2i, while the di-phosphorylated forms increased in Cdk8/19i (Fig. 4c). Although these metabolites can be diverted for multiple processes, we noticed that enzymes involved in nucleotides biosynthesis (Pgls, Gart, and Atic) were downregulated only in 2i (Supplementary Data 1), which could explain the differences in nucleotide pools. These findings indicate that some important aspects of nucleotide metabolism also differ between 2i and Cdk8/19i.

Another important difference between 2i and Cdk8/19i was in their DNA methylation levels. Despite their similarities, Cdk8/19i cells do not phenocopy the hypomethylated genome of 2i[3]. Hence, these two culture conditions might help understand the mechanisms of DNA methylation in pluripotency. Nnmt acts as a methyl sink in human naïve ESCs through methylation of nicotinamide (NAM), making SAM unavailable for other methylation reactions[9]. We found Nnmt mRNA expression increased 16-fold in 2i, while it diminished in Cdk8/19i (Fig. 4d). Consistent with these changes, our data showed low SAM levels in 2i. Conversely, a high SAM/SAH ratio, which activates methyl-transferases, was found in Cdk8/19i (Fig. 4d). These results suggest that SAM availability could in part determine DNA methylation in ESCs. In addition, DNA methylation depends on the balance between de novo methylation, maintenance methylation, and active demethylation, and all three processes have been implicated in the loss of methylation in 2i[7,38]. In our data, Dnmt3a and Dnmt3l were among the most downregulated

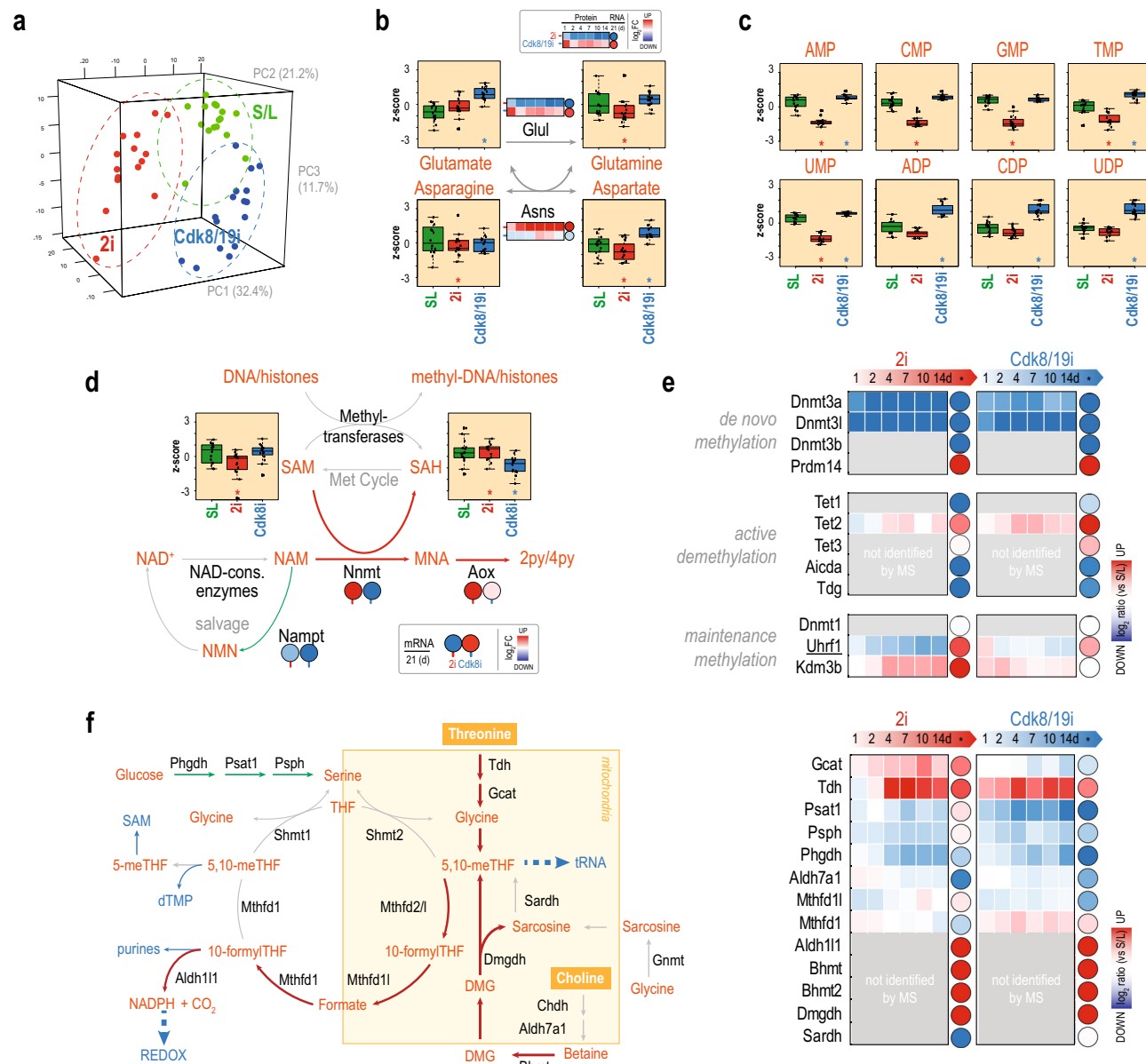

**Fig. 4 2i and Cdk8/19i impose specific metabolic signatures. a** Principal component analysis of the metabolomes of mESCs grown in S/L, 2i, or Cdk8/19i. **b** Glutamate, glutamine, asparagine, and aspartate levels found in S/L, 2i, and Cdk8/19i. The protein (squares) and RNA (circles) log2 ratios relative to S/L are indicated for Glul and Asns enzymes. **c** Mono-phosphorylated and di-phosphorylated nucleotides levels in S/L, 2i, and Cdk8/19i. **d** Schematic of methylation reactions. SAH and SAM levels measured in S/L, 2i, and Cdk8/19i culture conditions. mRNA log2 ratios of Nnmt in 2i and Cdk8/19i and related enzymes. **e** Expression log2 changes of enzymes involved in DNA (de)methylation. **f** Schematic of the one carbon metabolism and related enzyme levels. When available, protein (squares) and mRNA (circles) ratios are shown. In all boxplots (**b–d**), statistically significant changes using a moderated *t*-test (limma, two-sided, FDR<5%) relative to reference S/L are indicated with an asterisk (*p*-value <0.05). In **b–d**, boxplots represent values from four independent cell lines (with four biological replicates each) in S/L. 2i and Cdk8/19i ($n = 16$). Boxplots show medians and limits indicate the 25th and 75th percentiles as determined by R software; whiskers extend 1.5 times the interquartile range from the 25th and 75th percentiles, outliers are represented by dots. Source data for figures 4a–f are provided as Source Data file.

proteins in 2i, while their upstream negative regulator Prdm14 increased (Fig. 4e). Remarkably, Cdk8/19i recapitulated the same changes (Fig. 4e). On the other hand, TET dioxygenases, Aicda and Tdg also exhibited similar regulation (Fig. 4e). Recently, impairment of DNA methylation maintenance was proposed as the main cause of hypomethylation in 2i[39]. We found that Uhrf1 protein levels, but not its mRNA, rapidly decreased in 2i but not in Cdk8/19i (Fig. 4e). Kdm3b, a H3K9 demethylase was also found increased only in response to 2i. Although we cannot rule out additional factors such as interactions, PTMs and metabolites

regulating the activity of de novo methyltransferases and active demethylases, our data seem at least to support the model where Uhrf1 degradation, concomitant with Kdm3b-mediated loss of H3K9me2, impede Dnmt1 recruitment, leading to a passive loss of 5mC in 2i[39]. Moreover, the high levels of SAM together with no change in Uhrf1 protein in Cdk8/19i likely explain how Cdk8/19i does not affect global DNA methylation levels[3].

Despite the differences in the above metabolic pathways, our data also showed that 2i and Cdk8/19i caused similar alterations in other routes, such as the one carbon metabolism (OCM).

While serine biosynthesis proteins (Phgdh, Psat, and Psph) were repressed in 2i and Cdk8/19i, Tdh (involved in the catabolism of threonine into glycine) was upregulated several fold[40] (Fig. 4f). Further, choline degradation products are another important source of 1C units. We found Bhmt, Bhmt2, and Dmgdh mRNAs (all involved in the re-methylation of homocysteine in the liver), were 30-fold increased in both media (Fig. 4f and Supplementary Fig. 4D). Thus, these findings suggest that, besides threonine, naïve ESCs may incorporate 1C units by substantially mobilizing choline. Interestingly, both sources converge into 5,10-meTHF which is essential to produce the taurinomethyluridine base of mitochondrial tRNAs[41] (see "Discussion" section).

**Naïve cells upregulate proteins involved in fatty acids beta-oxidation.** Hif1a promotes a glycolytic metabolism in EpiSCs[6]. Cdk8 kinase activity is required for induction of many Hif1a targets and supports glycolysis in cancer cells[42,43]. In concordance, our proteome data showed multiple Hif1a targets involved in glycolysis were downregulated by both 2i and Cdk8/19i (Fig. 5a). Moreover, several glycolytic metabolites were reduced in 2i or Cdk8/19i (Fig. 5a and Supplementary Data 4). Furthermore, we found that both inhibitors increased most of the TCA cycle and ETC enzymes (Fig. 5b). TCA intermediates however showed little

changes. Together, these results suggest that naïve cells may re-balance the bivalent metabolism by enhancing mitochondrial respiration while diminishing glycolysis.

The major change in our metabolomic data involved the lipid content. 36% (2i) and 42% (Cdk8/19i) of the downregulated metabolites belonged to lipid metabolism. Importantly, 30 of them decreased in both inhibitors ($p.val = 3E-15$) indicating that both drugs induced a similar lipidomic signature (Fig. 5c). Interestingly, 50% of the upregulated metabolites in 2i were also lipids, including glycerophospholipids and sphingomyelins (Supplementary Data 4). Lipid metabolism is emerging as an important player in pluripotency but studies have reported diverging results on the role of fatty acid oxidation[9] and de novo lipogenesis[44]. Here, the overall decrease in lipids found in 2i and Cdk8/19i suggests high beta-oxidation levels and/or reduced lipid synthesis. We found that lipid catabolism and beta-oxidation pathways including Cpt1a and Slc25a20 from the carnitine transport system were upregulated (Fig. 5d, e). Peroxisomal enzymes like Acox3 and Decr2 also increased, indicative of oxidation of very long fatty acids as well (Fig. 5d). Furthermore, during mouse development, the expression of beta-oxidation enzymes is highest in the ICM and pre-implantation[45] and their protein levels are rapidly down-regulated as cell transit to primed pluripotency[10]. Although Slc25a1 and Fasn were found slightly

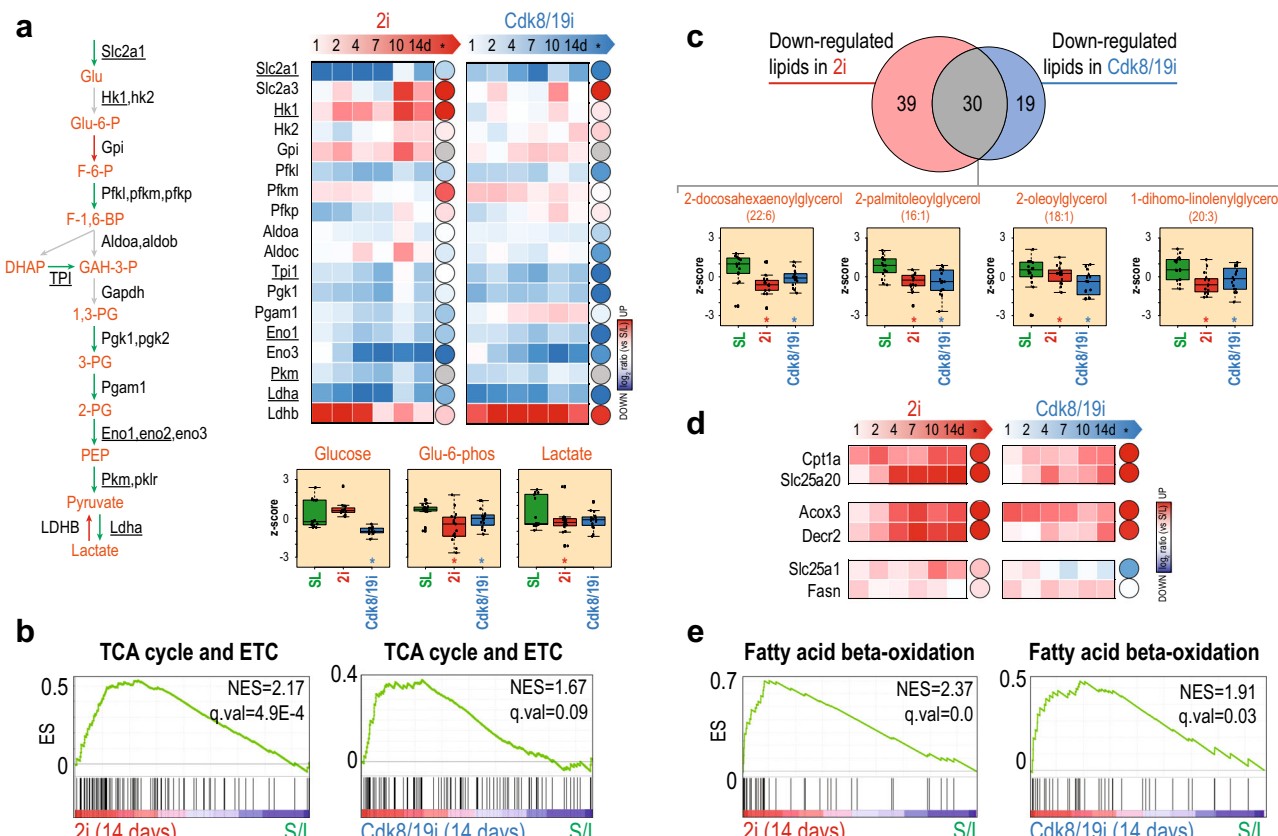

**Fig. 5 Naïve cells possess an enhanced oxidative phosphorylation and beta-oxidation. a** Schematic of glycolysis reactions and related enzyme levels (relative to S/L) in 2i and Cdk8/19i. Protein (squares) and mRNA (circles) log2 ratios are shown. Hif1a targets as described by Galbraith et al.[42] are underlined. Below, glycolytic metabolites found altered in 2i or Cdk8/19i (relative S/L) using a moderated *t*-test (limma, two-sided, FDR<5%) are indicated with an asterisk (*p*-value <0.05). **b** Upregulation of proteins related to TCA and ETC at 14 days of treatment with 2i or Cdk8/19i shown by GSEA analysis. **c** Overlap between the lipids down-regulated in 2i and Cdk8/19i conditions (relative to S/L). Some examples of down-regulated lipids are shown below. **d** Log2 ratios of key enzymes involved in the carnitine transport system, peroxisomal oxidation of lipids and fatty acid synthesis. **e** GSEA shows the upregulation of fatty acid beta-oxidation proteins at 14 days of treatment with 2i or Cdk8/19i. In **a** and **c**, boxplots represent values from four independent cell lines (with four biological replicates each) in S/L, 2i and Cdk8/19i (*n* = 16). Boxplot show medians and limits indicate the 25th and 75th percentiles as determined by R software; whiskers extend 1.5 times the interquartile range from the 25th and 75th percentiles, outliers are represented by dots. Source data for Figs. 5a, c and d are provided as Source Data file.

upregulated (Fig. 5D), we did not find any GO term that would support increased lipid biosynthesis (Supplementary Data 2). Collectively, these results might imply that mitochondria in the naïve state are metabolically active and characterized by high levels of beta-oxidation.

**Phosphoproteomics confirms the selectivity of Mek/Gsk3i and Cdk8/19i.** The similar proteomic and metabolomic changes of 2i and Cdk8/19i prompted us to investigate the immediate effects of these drugs using phosphoproteomics. Four S/L mESC lines were supplemented with 2i or Cdk8/19i and collected after 0.5, 1, 2, and 6 h (Fig. 1a, Supplementary Fig. 5A, and Supplementary Note 3). Samples were distributed across four TMT-11plexes, and purified phosphopeptides were analyzed by LC-MS/MS, enabling the quantification of 14,276 p-sites in all 44 conditions (Supplementary Fig. 5B). PCA revealed a clear separation of 2i and Cdk8/19i phosphoproteomes from S/L, indicating a profound alteration in the phosphorylation landscape of mESCs immediately upon addition of these kinase inhibitors (Fig. 6a). 4261 and 5732 p-sites were significantly regulated after 0.5 h of 2i or Cdk8/19i respectively (Fig. 6b and Supplementary Data 5). Strikingly, 40% of these changes were common to both inhibitors. Similar overlaps were found at 1, 2, and 6 h (Fig. 6b). Clustering classified p-sites into 14 temporal profiles including six clusters exhibiting alike kinetics in both inhibitors (Fig. 6c and Supplementary Fig. 5C). These results suggested that 2i and Cdk8/19i re-wired phosphorylation networks in a similar manner. Next, we examined the status of the 43 phosphatases and 137 kinases identified in our data, and found that several of them displayed comparable phosphorylation regulation in 2i and Cdk8/19i, which could explain the partial convergence in their phosphoproteomes. For instance, S1261-T1262 from mTOR were heavily downregulated, suggesting decreased kinase activity[46] (Fig. 6d and Supplementary Fig. 5D). Consequently, Ulk1 showed dephosphorylation of several residues including S637 (mTOR site), an event that promotes an acute autophagy induction[47]. Furthermore, our data showed inhibition of kinases involved in protein synthesis such as S6K (S394, mTOR site) and Eef2k (T347, S6K site) (Fig. 6d and Supplementary Fig. 5D). In addition, 2i and Cdk8/19i-induced phospho-changes showed widespread effects on nuclear functions (Supplementary Fig. 6 and Supplementary Data 6). For instance, we found cell cycle proteins similarly regulated in both inhibitors (cluster #2, $q$.val = 4E−11). This included hypo-phosphorylated RB that, together with the increased expression of p21/p27, confirmed the elongated G1 phase typical of naïve ESCs[48]. Interestingly, our data also suggested that naïve cells are partially relieved from the G2/M checkpoint through inhibition of Chek1: phosphorylation of its catalytic site (S317) significantly diminished, and its motif was enriched among the downregulated p-sites ($q$.val=1.6E−10) (Supplementary Fig. 7 and Supplementary Data 7), which included known Chek1 targets.

Despite the above similarities, we noted some important differences in the phosphoproteome response to Cdk8/19i and 2i. The two most differential p-sites between 2i and Cdk8/19i belonged to Mek (Fig. 6f). The PD0325901 structure does not disrupt the phosphorylation of the Mek activation loop, which was consequently relieved from the negative feedback of the pathway, resulting in the accumulation of phosphorylated Mek in S222/S226[49] (Fig. 6g). As expected, Mek inhibition reduced Erk phosphorylation at Y185 of the TEY motif (Fig. 6g). Likewise, we found several changes in p-sites of Raf proteins, altogether confirming the potent inhibition of the Fgf/Mek/Erk pathway, within the first 6 h of exposure to the PD0325901 inhibitor. Meanwhile, the Gsk3 inhibitor, CHIR99021, led to progressive

stabilization of beta-catenin protein levels and, consequently, upregulation of Axin2 and Cdx1[50] (Supplementary Data 9).

Cdk8 kinase activity is less understood and its effects are cell context dependent. Among the few substrates known for Cdk8 is histone H3[51], which showed decreased phosphorylation in S29 in Cdk8/19i. Recently, Poss et al.[52] identified 64 putative Cdk8/19 phospho-targets in human cancer cells using a different Cdk8/19 inhibitor, cortistatin A[52]. Despite several significant experimental differences between both data sets, we identified 13 of those substrates in our data, of which, five showed diminished phosphorylation in response to Cdk8/19i (Supplementary Data 5). Of note, these proteins were tightly linked to transcription, including the regulatory subunit of the kinase module Med13l, Brd9, Nab2, Znf609, and Huwe1.

To understand the differential response of 2i and Cdk8/19i, we focused on Gsk3. Since Gsk3 is a constitutively active kinase targeting S/T at four residues upstream of a pre-phosphorylated priming site, we checked in our data for doubly-phosphorylated peptides bearing such motif. Indeed, this Gsk3 motif was highly enriched among the phosphopeptides decreasing in 2i (cluster #6, $p$.val = 9.37E−18), and almost half of those peptides were co-phosphorylated in the priming site (Supplementary Fig. 7 and Supplementary Data 8). Due to diminished Gsk3 activity in 2i, we reasoned that the priming mono-phosphorylated peptide could remain unaltered or even increase, providing further evidence of its identity as a Gsk3 substrate and reveal information on its stoichiometry. For instance, the di-phosphorylated T58-S62 peptide of Myc decreased in 2i, with a concomitant increase of the priming mono-phosphorylated S62 (Fig. 6h). Notably, our data identified multiple potential Gsk3 targets including S324/T328 in Ezh2 (Supplementary Fig. 8). Interestingly, Gsk3 has been reported in cancer cells to phosphorylate these Ezh2 residues in the cytosol[53] and mounting evidence shows that cytosolic Ezh2 has non-canonical roles through methylation of non-histone proteins[54]. Thus, despite the known role of Ezh2 in the naïve state as part of the PRC2 complex by protecting 2i-ESCs from primed-like features[14], our data suggest that Ezh2 might also have additional functions in a PRC2-independent manner.

**An interplay of downstream transcriptional effectors are regulated in 2i.** Among the proteins showing the highest differences in phosphorylation between 2i and Cdk8/19i we found Erf (Supplementary Fig. 9A). Erf is a direct sensor of Erk activity, with links to pluripotency[55]. Our data shows that in 2i, Mek-Erk signaling inhibition leads to rapid dephosphorylation of Erf at S20/21 (Fig. 7a and Supplementary Fig. 9B), which, as corroborated by immunofluorescence, facilitates its stabilization and nuclear translocation[55–57] (Fig. 7b). In contrast, Cdk8/19i did not produce any effect on Erf phosphorylation, nor changes in its localization (Fig. 7a, b and Supplementary Fig. 9B), indicating that Cdk8/19i had no impact on this ERK downstream target. Nuclear Erf is mainly a transcriptional repressor but it can also activate gene expression[55]. To assess the contribution of Erf in naïve pluripotency, we mapped Erf targets, as reported by Mayor-Ruiz et al.[55], in our data. Genes where Erf has been shown to act as a repressor were downregulated only in 2i, whereas those where Erf acts as a transactivator were predominantly increased (Fig. 7c and Supplementary Fig. 9D). These transcriptional changes occurred within 24 h of 2i, suggesting that Erf might be an early downstream effector of 2i. Among the Erf-repressed genes as reported in Mayor-Ruiz et al.[55], there are negative regulators of the Erk pathway ($q$.val = 5E−3) including DUSPs and SPRYs but also genes from the primed state such as Otx2, Utf1, Sox3, Lin28b, and Lef1[31,58,59] all

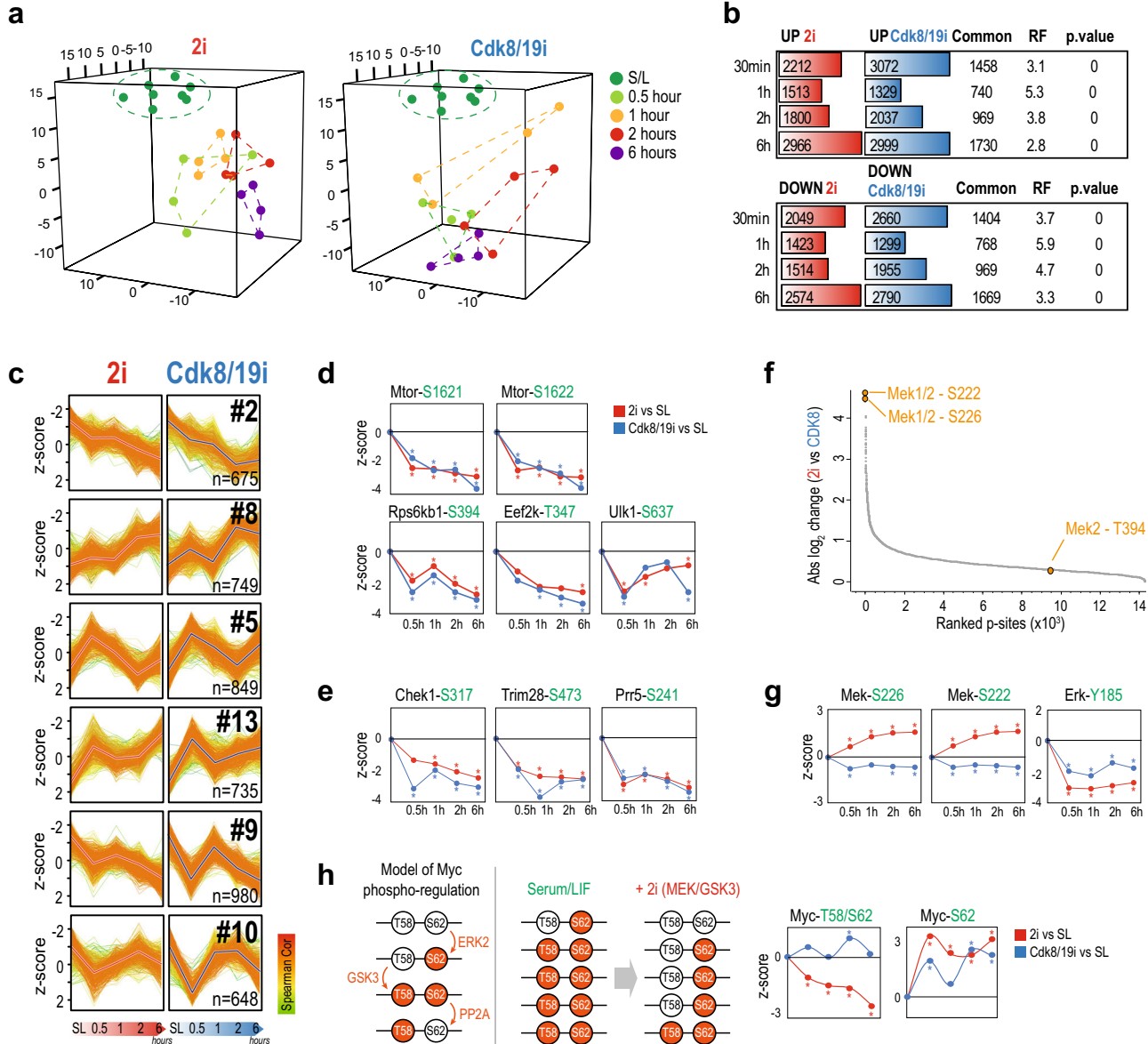

**Fig. 6 Time-resolved phospho-proteomics in response to 2i or Cdk8/19i. a** Principal component analysis of the phospho-proteomes of cells treated with 2i or Cdk8/19i at different time points. **b** Differentially regulated p-sites using a moderated *t*-test (limma, two-sided, FDR < 5%) at the indicated time points. The number of common p-sites between 2i and Cdk8/19i is shown together with the corresponding *p*.values calculated with a hypergeometric test (RF indicates "Representation factor"). **c** Gap statistics and K-means classified the regulated phospho-proteome into 14 different clusters. Centroids are indicated with a thick red (2i) or blue (Cdk8/19i) line. Only six clusters are shown in the figure. The other 8 clusters are shown in Supplementary Fig. 5C. Color gradient in line plots indicates proximity to the centroid of the cluster. **d** Changes in p-sites of kinases involved in mTOR signaling in 2i or Cdk8/19i (*n* = 4 independent cell lines). Statistically significant changes using a moderated *t*-test (FDR < 5%) relative to reference S/L are indicated with an asterisk (*p*-value < 0.05). **e** Changes in phosphorylation of S317 regulatory site of Chek1 as well as two known Chek1 substrates (*n* = 4 independent cell lines). Statistically significant changes using a moderated *t*-test (FDR < 5%) relative to reference S/L are indicated with an asterisk (*p*-value < 0.05). **f** Ranked p-sites by their absolute difference between 2i and Cdk8/19i. Mek2 p-sites identified in our dataset are indicated. **g** Phosphorylation changes of Mek/Erk regulatory sites. Points indicate the average of *n* = 4 independent cell lines. Statistically significant changes using a moderated *t*-test (FDR < 5%) relative to reference S/L are indicated with an asterisk (*p*-value < 0.05). **h** Schematic model of Myc phospho-regulation in S/L and 2i. Changes in the T58/S62 di-phosphorylated Myc peptide and the S62 monophosphorylated are shown (*n* = 4 independent cell lines). Statistically significant changes using a moderated *t*-test (FDR < 5%) relative to reference S/L are indicated with an asterisk (*p*-value < 0.05). Source data for Fig. 6a are provided as Source Data file.

of which were downregulated in 2i (Fig. 7d). Furthermore, Erf-repressed genes tend to exhibit promoter proximal pausing of RNAPII (*p*.val = 2E−22) and overlapped with those activated by Zic3 (*p*.val = 1E−22), which promotes the transition to primed pluripotency[60] (Supplementary Fig. 9C), highlighting the potential role of Erf as a repressor in maintenance of the naïve

state. On the other hand, among the genes where Erf is reported to increase transcription[55], there are key naïve pluripotency factors including Prdm4 and Tbx3 that were upregulated in 2i (Fig. 7d). Similarly, we found up-regulation of other critical targets of pluripotency from the LIF-STAT3, Wnt and BMP4-SMAD signaling pathways (Fig. 7d).

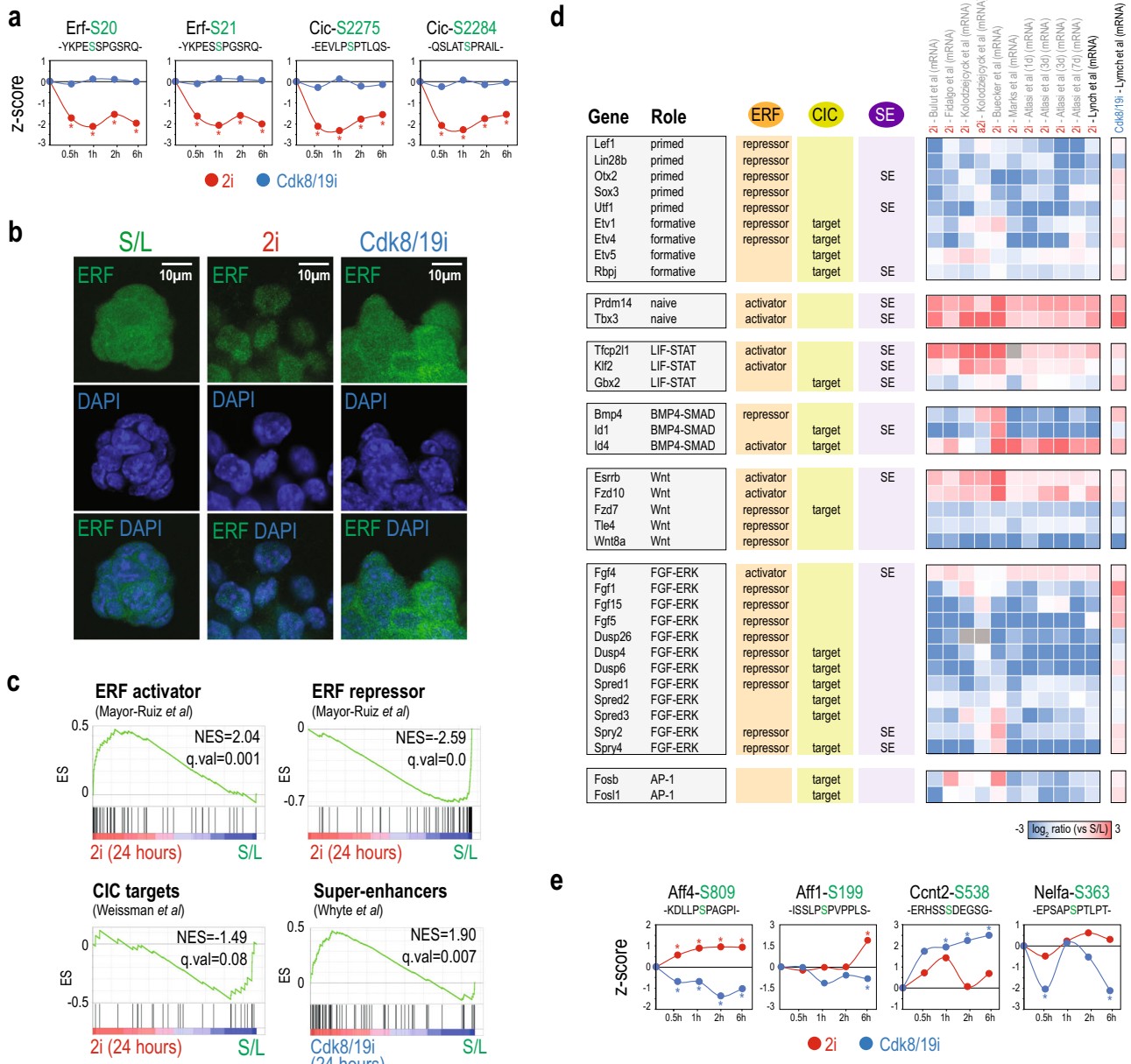

**Fig. 7 Phosphoproteomics identifies downstream proteins that stabilize the transcriptional naïve circuitry in 2i and Cdk8/19i. a** 2i but not Cdk8/19i reduces phosphorylation levels of Erf and Cic ($n = 4$ independent cell lines). Statistically significant changes using a moderated $t$-test (FDR < 5%) relative to reference S/L are indicated with an asterisk ($p$-value < 0.05). **b** Immunofluorescence analysis of ERF subcellular localization in mESCs in S/L o 2i or Cdk8/19i. Experiment was performed in duplicates. **c** GSEA results using gene sets extracted from the indicated sources. **d** Transcription factors and components of signaling pathways involved in pluripotency, indicating whether they are regulated by Erf (Mayor-Ruiz et al.[55]), Cic (Weissmann et al.[61]) and/or controlled by super-enhancers (Whyte et al.[68]). The mRNA log2 ratios are shown for Cdk8/19i and 2i (additional 2i data sets are also shown). **e** Phosphorylation levels ($n = 4$ independent cell lines) of several Cdk8/19 targets in response to 2i or Cdk8/19i. Statistically significant changes using a moderated $t$-test (FDR<5%) relative to reference S/L are indicated with an asterisk ($p$-value < 0.05).

Our phosphoproteomic data uncovered capicua (Cic) as another transcriptional target that was rapidly dephosphorylated only in 2i (Fig. 7a and Supplementary Fig. 9A, B). Similar to Erf, 2i impedes Erk-mediated nuclear export and/or degradation of Cic, leading to its nuclear accumulation and activation[61]. In agreement with its role as a repressor, we found that several Cic targets, as reported by Weissmann et al[61], were repressed in 2i. These include formative pluripotency genes[58], ERK negative regulators as well as AP-1 factors (Fig. 7c, d). Moreover, we also identified Cic as a potential substrate of Gsk3 (Supplementary Data 8), suggesting that both Mek-Erk and Gsk3 pathways might have synergistic downstream effects. Altogether, these data suggest that in 2i, ERKi,

and GSK3i converge in the stabilization of downstream factors that, probably in cooperation with other proteins, participate in: (i) repressing the transcriptional programs of development; (ii) ensuring the shutdown of the Erk pathway; and (iii) reinforcing the naïve circuitry.

**Activation of super-enhancer-associated genes key for pluripotency in Cdk8/19i.** Unlike 2i, inhibition of Cdk8/19 did not regulate the above transcriptional factors but recapitulated much of the expression program of 2i. To understand the mechanism of action of Cdk8/19i, we checked p-sites regulated under the effects

of this inhibitor but unaffected or with opposite kinetics in 2i (Supplementary Data 5). For instance, Mcm3, Claspn, and Ticrr were regulated by phosphorylation in response to Cdk8/19i, supporting a recent report that links Cdk8 with DNA replication[62]. In addition, we found Cdk8/19i-specific changes in Aebp2 and Jarid2 suggesting that Cdk8 may also regulate PRC2[63]. The implication of Cdk8 on epigenetics in our data was further evidenced by dephosphorylation of Dnmt3l[64]. However, given the main role of Cdk8 in transcription, we focused on proteins involved in this process. We found dephosphorylation in Aff1 and Aff4 from the super elongation complex in Cdk8/19i but not in 2i. Moreover, Ccnt2, the cyclin of the pTEFb, and Cdk13, involved in CTD phosphorylation and elongation, were also regulated only in Cdk8/19i (Fig. 7e). Therefore, these results indicate that Cdk8/19i may elicit part of its effects through proteins involved in the productive elongation phase of RNPII. Although Cdk8 can activate gene expression[42,65], it is mostly known for its repressive effect on transcription as a negative regulator of mediator[66]. Remarkably, Cdk8/19 inhibition with cortistatin A has been shown to activate genes that are under control of super-enhancers (SE) in AML[67]. To test whether a similar mechanism plays a role in the stabilization of naïve pluripotency by our Cdk8/19 inhibitor, we checked Cdk8/19i-dependent expression of genes that are transcriptionally-regulated by SE in S/L mESCs[68]. Indeed, we found that Cdk8/19i, but not 2i, showed a positive co-regulation of protein and RNA levels of SE-genes (Fig. 7c and Supplementary Fig 9D). Importantly, this effect was evident by day 1 of Cdk8/19i exposure. Among the SE-genes upregulated by Cdk8/19i there were naïve pluripotency factors including Tbx3, Klf2 Esrrb, Prdm14, and Tcfcp2l1 (Fig. 7d). Other SE-genes that showed significant expression included metabolic enzymes and cell surface molecules. Nevertheless, among the genes under control of SEs there were also transcription factors associated to formative and primed states that were consequently upregulated, including Otx2, Utf1, Rbpj, Pum1, and Tcf15 (Fig. 7d). Of note, most of these factors have been reported to be repressed by Erf and Cic[55,61] and were downregulated in 2i. Similar regulation was found for other SE genes such as SPRYs, Epha2, and Smarcd1 that were found repressed in 2i while activated in Cdk8/19i. Together, our results suggest a model in which Cdk8/19i regulate, among others, proteins involved in transcriptional processes such as elongation, probably leading to de-repression of SE-driven expression of key identity genes. These include components of the naïve transcriptional circuitry but also some factors from primed and formative pluripotency.

## Discussion

To better understand the regulatory principles underpinning the establishment of naïve pluripotency, we used MS to characterize two chemical approaches that target extracellular signaling (2i) and transcriptional machineries (Cdk8/19i). We propose a model in which 2i and Cdk8/19i result in a similar activation of naïve genes through different mechanisms. Although more factors might be involved, we showed that Erk inhibition results in the rapid activation of Cic and Erf (Fig. 8). The integration of our phosphoproteomic and transcriptomic data with published ChIP-seq data suggest that these downstream effectors might participate in the transcriptional response that activates the naïve program, while repressing the more developmentally advanced primed and formative states. Alternatively, a similar outcome is achieved by stimulation of key stem cell genes under the control of super-enhancers. Inhibition of Cdk8/19, a repressor of the Mediator complex, regulates proteins involved in processive transcription elongation. In Cdk8/19i, naïve but also some

primed and formative genes are activated (Fig. 8). Yet, mESCs grown in Cdk8/19i functionally resemble the 2i state[3] demonstrating the robustness of the self-organizing naïve network to destabilizing cues.

Once activated, the naïve circuitry entails the execution of a complex sequence of molecular events. We found that a feature of the naïve state is the upregulation of mitochondrial proteins, which seems to be partially controlled post-transcriptionally. Interestingly, these results contrast with a recent report that found that proteomic changes in the naïve state are largely driven by transcriptomic rewiring, with little contribution of other regulatory layers[12]. We have now re-examined these results and found evidences of post-transcriptional regulation in their data too (Supplementary Note 2). Genes regulated mainly at the translational level in Atlasi et al[12]. (RNA < RFP) were enriched in endoplasmic reticulum and plasma membrane (FDR = 0) but not in mitochondria. However, we found that mitochondrial proteins were indeed more upregulated at the protein than the mRNA ($p$.val = 1.2e−38) or RFP ($p$.val = 1.9e−31) levels. Moreover, in agreement with our results, divergences in mRNA-protein in Atlasi's data were coincident with the downregulation of Lin28, underscoring the known role of this RNAbp in post-transcriptional repression in mESCs[12]. Given that RFP is a direct reflection of translational activity, it is intriguing that the upregulation of mitochondrial protein levels (MS) was not detected in the RFP data. Multiple explanations may account for such discrepancy including differences in bulk vs. local RFP measurements and stalled/paused polysomes (Supplementary Note 2).

Interestingly, the upregulation of mitochondrial proteins may be facilitated by two additional events. Firstly, in line with prior reports[69–71], we found that either 2i or Cdk8/19i lead to de-phosphorylation of mTOR, possibly affecting catabolism/anabolism rates. It is well established that protein synthesis is repressed under stress conditions, where the translation machinery is diverted to selectively promote stress-response proteins[72]. A similar translational reprogramming might occur here: the substantial demand for translation of mitochondrial proteins to transition to the naïve state could be counterbalanced by tuning down global protein synthesis, as well as the recycling of cellular components through autophagy. Secondly, the over-expression of several enzymes that feed 1C units suggests that naïve cells may adapt their one carbon metabolism to cope with mitochondrial translation rates. Although this would require further examination of one-carbon fluxes by means of stable isotope tracers. Interestingly, both threonine and choline sources converge into 5,10-meTHF which is essential to produce the taurinomethyluridine base of mitochondrial tRNAs[41]. Therefore, it is our hypothesis that naïve ESCs might increase 5,10-meTHF levels to meet the demand of methylated mitochondrial-tRNAs needed for maintaining the high translation rates of mitochondrially-encoded proteins. Of note, these enzymes are repressed as cells progress to primed pluripotency[10], and they exhibit specific expression in the morula and ICM[2], further supporting their roles in pluripotency.

Despite their similarities, 2i and Cdk8/19i showed differences in several metabolic pathways. The upregulation of Nnmt in 2i diverts SAM from methylation reactions[9]. In contrast, Nnmt levels remain basal in Cdk8/19i, consistent with their lack of hypomethylation[3]. Another contributing factor could be the 2i-dependent degradation of Uhrf1 that impedes the recruitment of Dnmt1 leading to a passive loss of methylated bases during replication[39]. Although low levels of DNA methylation are considered a hallmark of naïve pluripotency given its resemblance with the demethylation wave of the ICM, it has been reported that prolonged culture in 2i can have deleterious effects on the

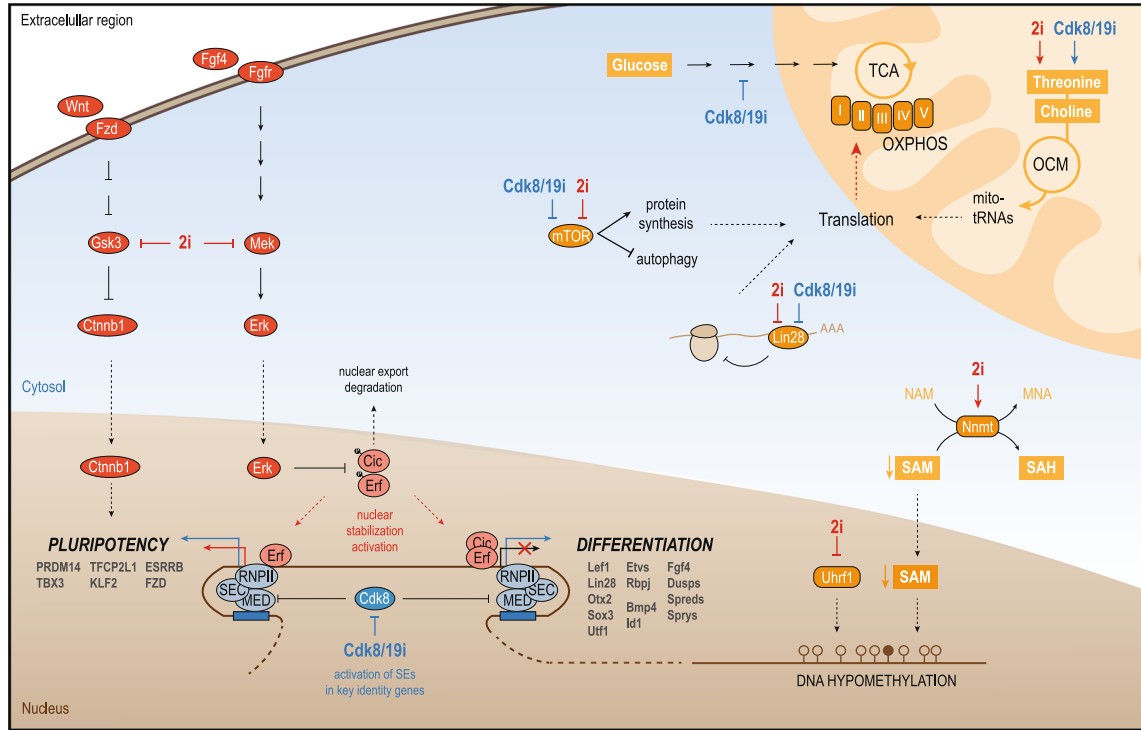

**Fig. 8 Working model for 2i and Cdk8/19i -mediated mechanisms involved in the stabilization of naïve pluripotency.** 2i-dependent inhibition of Mek/ Erk prevents phosphorylation of Erf and Cic, stabilizing both factors in the nucleus. These transcriptional effectors might participate in the activation of genes involved in naïve pluripotency whereas they repress factors involved in primed/formative states. Cdk8/19i affects the phosphorylation of proteins involved in elongation and transcription machineries such as super elongation complex (SEC), RNA pol II (RNPII) and Mediator (MED). This might lead to rapid activation of genes controlled by super-enhancers. These genes include factors that stabilize the naïve state but also some factors from the primed/ formative states which are also upregulated in response to Cdk8/19i. Upon activation of the naïve circuitry in 2i and Cdk8/19i, several events might support the enhanced mitochondrial OXPHOS capacity of these cells. Degradation of Lin28a might relieve target mRNAs from its repression effects and, among others, mitochondrial proteins become upregulated. The balance of autophagy and protein synthesis may sustain the translation of such mitochondrial components. In addition, re-wiring of mitochondrial sources for one carbon units may also sustain the translation of mitochondrial-encoded proteins. On the other hand, the availability of SAM levels (by NNMT upregulation) together with the impairment of DNA methylation (by Uhrf1 degradation) contribute to the hypomethylated DNA typical of 2i.

chromosome integrity of mESCs[8]. In this scenario, Cdk8/19i would be an attractive alternative, as these cells exhibit similar functional capacities to 2i, while methylation is stable and not subject to progressive loss[3].

In our previous work, we have shown that Cdk8/19i also shifts human ESCs towards the naïve state[3], hence, some of the events described here might play a role in this system too. Moreover, our data might be valuable in other areas such as cancer. Inhibition of Cdk8 reduces cell proliferation in AML through activation of tumor suppressors[67]. In colorectal cancer cells however, Cdk8 acts as an oncogene[43]. Here, we show that Cdk8/19i leads to phosphorylation changes in hundreds of proteins, which may help in understanding the context-dependent functions of these kinases. While the primary function of Cdk8/19 resides in the nucleus, and potential off-targets cannot be neglected, Cdk8/19 can indeed function independent of core Mediator[51] and it has been recently found in the cytoplasm[73], suggesting that the target spectrum of these kinases may be larger than anticipated. On the other hand, the Erk pathway involves multistep reactions and feedback loops, that generate various behaviors. The PD0325901 used in 2i, and many other Mek inhibitors, do not disrupt the phosphorylation of the Mek1/2 activation loop, which can explain the rebound in phosphorylated Erk1/2 and pathway output that is observed with this type of inhibitors, notably in Ras-driven cancers. Indeed, one of the downstream effectors identified here, Cic, has been shown to modulate sensitivity to Mek-inhibitors[74].

Thus, our data might be used to identify potential vulnerabilities and resistance mechanisms to such compounds.

## Methods
More detailed descriptions of the methods employed in the current study is included as Supplementary Methods in the supplementary information.

**Mouse cells and culture conditions**. The experimental design consisted of four different mouse embryonic stem cell lines: TNGA, TON, ZS and BL6. Wild-type ES cells were derived at the Transgenic Mouse Unit of CNIO from E3.5 C57BL6 blastocysts (BL6), or mixed background C57BL6/129 (V6.4) blastocysts. Nanog-GFP knock-in mouse ES cells (TNGA, TON) and were shared by the laboratory of Austin Smith; The ZS mouse ES line was a 2C-reporter shared by the laboratory of Minoru Ko. Mouse ES cells were routinely cultured on gelatin-coated plates in a base media of "Serum/LIF" (15% FBS), in DMEM (high glucose) basal media, with LIF (1000 Units/ml), non-essential amino acids, glutamax and β-mercaptoethanol plus antibiotics. Where used with mouse PSC, the "2i" two-inhibitor cocktail comprised 1 μM Mek-inhibitor (PD0325901, Axon Medchem, #1408) plus 3 μM GSK3β-inhibitor (CHIR 99021, Axon Medchem #1386). The Cdk8/19i was supplemented at 1.1 μM. Cultures were routinely tested for mycoplasma. Cell pellets were collected by trypsinization at different time points after treatment with 2i or Cdk8/19i: 30 min, 1, 2, and 6 h, 1, 2, 4, 7, 10 and 14 days of treatment and washed with PBS and preserved at −80 °C for further analysis.

**Sample processing and LC-MSMS analysis for proteomics**. Four mouse embryonic stem cell lines were used: TNGA, TON, ZS, and BL6. Each cell line was employed as a biological replicate, so the overall experiment consisted in four biological replicates. For time course full proteome analysis cells were lysed using 7 M urea, 2 M thiourea, 50 mM Hepes, 1:1000 (v/v) of benzonase and 1:100 (v/v) of HaltTM phosphatase and protease inhibitor cocktail 100×. Cell lysates were

homogenized by sonication and cleared by centrifugation. Protein concentration was measured with Qubit® Protein Assay Kit. One hundred and ten microgram of each lysate (except for control conditions for which 220 µg were used) were digested using the filter aided sample preparation (FASP) method. A first digestion using Lys-C (1:50 w/w, Wako Pure Chemical Industries) for 4 h was followed by a dilution 8-fold in 50 mM TEAB and a subsequent digestion with trypsin (1:100 w/w, Promega) overnight at 37 °C. Labeling was performed using the iTRAQ® Reagent 8plex kit (AB Sciex). Labeled samples were combined and cleaned-up with C18 Sep-Pack. Eluate was dried and dissolved in 10 mM of $NH_4OH$ for subsequent fractionation by high pH reversed phase chromatography. Collected fractions were dried and dissolved in 50 µl of 1% FA for LC-MS/MS analysis. The Impact (Bruker Daltonics) was coupled online to a nanoLC Ultra system (Eksigent). 7.5 µl of each fraction were loaded onto a reversed-phase C18, 5 µm, 0.1 × 20 mm trapping column (NanoSeparations) and washed for 15 min at 2.5 µl/min with 0.1% FA. The peptides were eluted at a flow rate of 250 nl/min onto a home-made analytical column packed with ReproSil-Pur C18-AQ beads, 1.9 µm, 75 µm × 50 cm, heated to 45 °C. Solvent A was 4% ACN in 0.1% FA and Solvent B CH3CN in 0.1% FA. The MS acquisition time used for each sample was 180 min. The Q-q-TOF Impact was operated in a data dependent mode. The 20 most abundant isotope patterns exceeding a threshold of 5000 counts and with charge ≥2 and $m/z$ > 350 from the survey scan were sequentially isolated and fragmented in the collision cell by collision induced dissociation (CID).

For time course phosphoprotome profiling cells were lysed using 5% SDS and digested following the S-trap protocol. Samples were labeled using TMT11plex reagent following manufacturer instructions. Labeling reaction was stopped by addition of 1% hydroxylamine; then samples were combined and desalted using C18 Sep-Pack. To perform phosphopeptide enrichment, peptides were dissolved in 80% CH3CN and 6% TFA. Titanium dioxide ($TiO_2$) beads were prepared at 60 µg beads/µl of DHB solution (20 mg/ml DHB in 80% CH3CN 6% TFA). $TiO_2$ beads were added to the sample in a ratio 1:2 (Sample:$TiO_2$) and incubated for 15 minutes. Supernatant was used for a second binding with half the amount of beads. Beads from the first and second TiO2 binding were transfer to separate C8-tips and washed. Peptides were eluted with 25 µl of 5% $NH_4OH$ and 25 µl of 10% $NH_4OH$ 25% CH3CN. Eluate from the second $TiO_2$ binding was resuspended in 22 µl 5% FA for LC-MS/MS analysis. Eluate from the first binding was fractionated with high pH reverse phase microcolumns. Sample was loaded into the tips thrice and the flow-through was collected to a vial. Next, 50 µl of phase A (20 mM $NH_4OH$) was loaded and collected as the flow-through. Peptides were sequentially eluted increasing the percentage of Buffer B (20 mM $NH_3$ in CH3CN). Samples were dissolved in 22 µl of 5% FA for subsequent LC-MS/MS analysis. Three replicates were run for each sample. The experiments were performed on an Ultimate 3000 RSL nano LC system (Thermo Scientific) coupled to a Q Exactive HF-X mass spectrometer (Thermo Scientific) equipped with an EASY-spray ion source (Thermo Scientific). The peptides were eluted from an Easy-Spray Column (75 µm i.d. × 50 cm) packed with PepMap RSLC C18 2 µm by application of a binary gradient consisting of 0.1% FA (buffer A) and 100% ACN in 0.1% FA (buffer B), with a flow rate of 250 nl/min during 90 min. The column was operated at a constant temperature of 45 °C. The MS survey was performed in the Orbitrap for a $m/z$ range between 350 and 1500 $m/z$. The resolution was set to 60,000 FWHM at $m/z$ 200. The 15 most abundant isotope patterns with charge ≥2 and <6 from the survey scan were selected with an isolation window of 1 $m/z$ and fragmented in the HCD collision cell. Normalized collision energy was set to 35. The resulting fragments were detected in the Orbitrap for a $m/z$ range between 100 and 2000 $m/z$ with a resolution of 45,000 FWHM at $m/z$ 200. The maximum ion injection times for the survey scan and the MS/MS scans were 45 ms and 80 ms, respectively and the ion target values were set to 3e6 and 5e4, respectively for each scan mode. Raw data was acquired using Xcalibur (Q-Exactive software Tune version 2.9, Thermo).

**Proteomics raw data processing and statistical analysis**. Proteome raw files were analyzed using MaxQuant 1.5.3.30 and 1.6.0.16 for full proteome and phosphoproteome datasets, respectively, against a Mus musculus database (Uni-ProtKB/Swiss-Prot, 43,539 sequences). For total proteome sample quantification type was set to iTRAQ8plex, whilst for phospho-proteome it was set to TMT11plex. Carbamidomethylation of cysteine was included as fixed modification and oxidation of methionine and acetylation of protein N-terminal were included as variable modifications. Phosphorylation of serine, threonine, and tyrosine was also included as variable modification in the phosphoproteomics dataset. Peptides and proteins were filtered at 1% FDR. For protein assessment (FDR < 1%) in MaxQuant, at least one unique peptide was required for identification. Only unique peptides were used for protein quantification. Reverse, only identified by site and proteins or phospho-sites that did not have reporter intensity in all channels were discarded for further analysis. Reporter intensities for the full proteome time course were extracted from "proteingroups.txt" table, and the reporter intensities for the phosphorylation sites were extracted from "Phospho(STY)Sites.txt" table. Data was further processed using R and Perseus. Reporter intensities were transformed to log2 and normalized within each experiment using "normalizecyclicloess" function from limma package in R. In order to integrate the independent experiments a further interexperiment normalization was conducted using ComBat function from

sva package in R.Statistical determination of differential proteins, and phosphorylation sites at each time point versus Serum/LIF initial conditions was performed using a two-sided limma approach implemented in Prostar (version 1.12.4). We set a minimum fold change of 0.1 (log2) and a maximum $p$-value for each condition. The $p$-value threshold was adjusted by Benjamini–Hochberg correction to a 5% FDR.

**Sample processing and LC-MSMS analysis for metabolomics**. Four mouse embryonic stem cell lines were used: TNGA, V6.4, ZS, and BL6. Metabolomic profiling was performed in collaboration with Metabolon Inc. Samples were prepared using the automated MicroLab STAR® system from Hamilton Company. Several recovery standards were added prior to the first step in the extraction process for QC purposes. To remove protein, dissociate small molecules bound to protein or trapped in the precipitated protein matrix, and to recover chemically diverse metabolites, proteins were precipitated with methanol under vigorous shaking for 2 min (Glen Mills GenoGrinder 2000) followed by centrifugation. The resulting extract was divided into five fractions: two for analysis by two separate reverse phase (RP)/UPLC-MS/MS methods with positive ion mode electrospray ionization (ESI), one for analysis by RP/UPLC-MS/MS with negative ion mode ESI, one for analysis by HILIC/UPLC-MS/MS with negative ion mode ESI, and one sample was reserved for backup. Samples were placed briefly on a TurboVap® (Zymark) to remove the organic solvent.

**Metabolite quantification, data normalization and statistical analysis**. Raw data was extracted, peak-identified, and QC processed using Metabolon's hardware and software. Compounds were identified by comparison to library entries of purified standards or recurrent unknown entities. Metabolon maintains a library based on authenticated standards that contains the retention time/index (RI), mass to charge ratio ($m/z$), and chromatographic data (including MS/MS spectral data) on all molecules present in the library. Furthermore, biochemical identifications are based on three criteria: retention index within a narrow RI window of the proposed identification, accurate mass match to the library $+/-$ 10 ppm, and the MS/MS forward and reverse scores between the experimental data and authentic standards. The MS/MS scores are based on a comparison of the ions present in the experimental spectrum to the ions present in the library spectrum. Peaks were quantified using area-under-the-curve. To account for differences in metabolite levels due to differences in the amount of material present in each sample, we have applied an additional normalization factor based on the total protein as determined by Bradford assay. One of the samples was discarded in this step due to the high presence of missing values (cell line V6.4, treatment 2i, and replicate 2). Afterwards, data processing was performed within Prostar R package (version 1.12.16). Metabolites that were not quantified in at least 75% of the samples in one treatment were discarded. Data was transformed to log2 scale and missing values were imputed using Imp4p for values missing at random, and with LAPALA with upper bound 2.5 for values missing in an entire condition. Once missing values were imputed, data was normalized by quantile centering to the median. Finally, with these data we performed differential analysis using limma (v3.36.1) with a design that account for biological replicates of the same cell line, with $p$-value correction of 5% FDR.

**Z-scoring, temporal trend classification, and boxplot representation**. Z-score or mean-centering normalization was applied to normalize protein and phosphosite intensities. Z-score was applied after collapsing all replicates to their average at each time point. Next, using the proteins differentially expressed at least at one time point in our treatment we calculated the optimal number of clusters using the "clusGap" in R. Based on that result, k-means clustering was performed using the function "cluster_analysis" from the library "multiClust" in R.

In all boxplot representations center lines show the medians; box limits indicate the 25th and 75th percentiles as determined by R software; whiskers extend 1.5 times the interquartile range from the 25th and 75th percentiles, outliers are represented by dots.

**Functional annotation**. Functional annotation of proteins was performed using either ClueGO, StringDB, or GSEA, using in all cases FDR control <5%. Potential kinase motifs were annotated using Perseus for class I phosphorylation sites. Fisher's exact test was performed to determine which motifs were significantly represented in each cluster. The background used for the enrichment analysis was comprised of all the class I phosphosites in the data (13,593 sites).

**Re-analysis of published transcriptomics data**. Processed data from the following selected publications was used in the current publication: Bulut-Karslioglu et al.[16] (GSE81285), Marks et al.[5] (GSE23943), Buecker et al.[59] (GSE56138), Kolodziejczyk et al.[75] (E-MTAB-2600), Fidalgo et al.[76] (GSE81045), and Lynch et al.[3] (GSE112208). Data was either downloaded directly from GEO repository or, if available, from Supplementary Material from the publication. Processed transcript instensities were log2 transformed and normalized by "Loess" (from limma package in R). Afterwards, differential genes between Naïve and Serum/LIF

conditions were estimated using the workflow "GEO2R" (https://www.ncbi.nlm.nih.gov/geo/geo2r/). Transcripts were considered significant if their fold change was above or below 1 (in log2 scale) and the *p*-value corrected by Benjamini–Hochberg was below 0.05. Upregulated and downregulated transcripts in 2i were used as gene-sets to perform rank-based enrichment analysis using GSEA algorithm. Ranked list of protein ratios in log2 at each time point of treatment was used as input for GSEA. GSEA was performed using default values. Normalized enrichment score was employed as scoring system.

**Reporting summary**. Further information on research design is available in the Nature Research Reporting Summary linked to this article.

## Data availability

The mass spectrometry proteomics data have been deposited to the ProteomeXchange Consortium via the PRIDE partner repository with the dataset identifier PXD018694. The mass spectrometry metabolomics data have been deposited to the Metabolights database with the identifier MTBLS301. Processed data from the following selected publications were used in the current publication: Bulut-Karslioglu et al.[16] (GSE81285), Marks et al.[5] (GSE23943), Buecker et al.[59] (GSE56138), Kolodziejczyk et al.[75] (E-MTAB-2600), Fidalgo et al.[76] (GSE81045), and Lynch et al.[3] (GSE112208). Source data are provided with this paper.

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

## Acknowledgements

We thank all members of the Proteomics Unit (CNIO), the Cellular Plasticity and Disease Group (IRB) and A. Efeyan (CNIO) for discussions. We thank O. Fernandez-Capetillo and M. Drosten for antibodies. We also thank the CNIO Experimental Therapeutics Programme for work on the CDK8/19i. I.C. is a recipient of predoctoral research contract from AGAUR. Work in the laboratory of M.S. was funded by the IRB and by the Spanish Ministry of Science co-funded by the European Regional Development Fund (ERDF) (SAF2017-82613-R), the European Research Council (ERC-2014-AdG/669622), and "laCaixa" Foundation. This work was supported by J.M. grant SAF2013-45504-R (MINECO). The CNIO Proteomics Unit belongs to ProteoRed, PRB3- ISCIII, supported by grant PT17/0019/0005. J.M. is supported by the Ramon y Cajal Programme (MINECO) RYC- 2012-10651. A.M.-V. is supported by BES-2014-070098 (MINECO).

## Author contributions

A.M.-V. performed all MS experiments and analysed the data. P.X.-E., F.G., and E.Z. contributed to the proteomic analyses. C.L. and I.C. generated all cell samples. A.M.-V., C.L., M.S., and J.M. planned all the experiments and wrote the manuscript. All authors contributed to and approved the final version of the manuscript.

## Competing interests

The authors declare no competing interests.
