## [Peer Review File · Nature Communications]

REVIEWER COMMENTS

Reviewer #1 (Remarks to the Author):

Within the manuscript “Dissection of two routes to naïve pluripotency using different kinase inhibitors” by Martinez-Val et al., the authors compare two strategies to induce naïve pluripotency in mouse ESCs: the canonical “2i” method, and a novel method using a Cdk8/19 inhibitor as very recently pioneered by the authors themselves (soon to be published in NCB). In particular, the authors use profiling of the (phospho)proteome and metabolome during transition towards naïve pluripotency to characterize potential regulatory mechanisms and key regulatory events associated with naïve pluripotency. I much enjoyed reading the manuscript, it is very well written with highly informative figures, clear in-depth analysis and a very straight storyline. Also, it is immediately clear that the authors are authorities on the subject, as they manage to provide a very comprehensive view on the event leading to naïve pluripotency, including (to me partly unexpected) roles for molecules such as mTOR, LIN28, CIC, ERF etc. Also, they combine a large range of previous observations into one holistic view. I noticed that the manuscript exclusively leans on (analysis of) the (phospho)proteome and metabolome profiling, and therefore this manuscript might also be labeled as a “data resource” paper. However, this directly shows two major weaknesses of the current paper: (i) it is highly specialized in a direct, rather descriptive comparison of two methods to induce naïve pluripotency in mESCs: are there any general principles from such extensive profiling to be conveyed towards a more general audience; (ii) none of the findings have been validated or (experimentally) followed-up. Any attempt to address these two issues would significantly improve the manuscript (in particular independent validation of the key findings).

Other major issues:

1) The authors compare two strategies to induce naïve pluripotency, but they never show clear validation of their timecourse using markers that have been extensively published for 2i and serum mESCs (and Cdk8/19i mESCs?). In particular, at the start of their manuscript, it would be highly informative (/essential) to show 5-mC and 5-hmC levels of the serum mESCs as being used, as well as after 14 days culturing in either 2i and Cdk8/19i, so to validate the system in their hands (at least in 2i this should show a very clear drop). This is also important as the authors later refer to 5-mC, but they never assayed it in their own system. Also, the authors should show a direct comparison of the dynamic changes of a few naïve and serum-specific markers between their study and published data to validate their system from the start on (rather than focusing on their own data only using z-scores).

2) I was very surprised to read that the authors used the 2i and Cdk8/19i on top of the serum, rather than replacing the serum. Notably, the general accepted and widely used naïve pluripotent state represents “2i” conditions without serum. In the conditions used in the paper, the mESCs are in a sort

of semi-naïve pluripotent state (which is why it is important to extensively validate this system, see point 1). It would be informative if the authors provide a rationale in the paper on why they

preferred this setup by adding the inhibitors on top of the serum. More importantly, I feel it is important that the authors validate their main findings in serum-free cultures. It would be highly interesting if the authors find some additional differences.

3) The analysis and representation of the phosphoproteomics is unclear to me. As far as I could find, the authors analyze phosphopeptides. But how do the authors discriminate between downregulation of a total protein versus specific changes in phospho-sites within that same protein. For example (but relevant for all such analysis): Are the phospho-sites of MTOR in Fig 6d specifically absent/ downregulated in naïve pluripotency, or is simply the whole MTOR protein downregulated. If the authors did not account for total protein changes, they should. Also, in that case, it would be preferred to show the proteome changes in parallel, or at least the fold-change in total protein after 1d of 2i or Cdk8/19i for which the authors have the data.

4) A recent manuscript by Atlasi et al. (referred to by the authors and also published in Nature Comm; PMID: 32238817) extensively mapped translational events in serum and 2i ESCs using ribosome profiling, which is very complementary to the current manuscript. Since part of the current manuscript also clearly focusses on the translational machinery (with various hypotheses and findings), the authors could very nicely incorporate the findings of Atlasi et al. to validate their findings (for example increased translation upon 2i or Cdk8/19i) or provide further evidence for some of their hypothesis regarding specific groups of proteins that are regulated at the translational level.

5) On page 11, line 357: “Hence, our data suggest that 2i might signal directly to EZH2 to regulate PRC2-independent processes.” However, there is clear evidence that EZH2 is also directly involved in maintaining naïve pluripotency as part of the PRC2 complex (PMID: 30472157). The authors might want to take this study into consideration regarding this specific issue.

Minor issues:

- For Fig 1F and similar figures further on, the authors should include within the figure the number of proteins included in each cluster.

- Page 3, line 77: A p.val of $2.7E-20$ seems very highly significant to me. On the other hand, I tend to agree that an overlap of 12 proteins out of ~1100 proteins does not seem significant, but this is not supported by such p-value?

- Page 6, line 182: “several targets of LIN28 showed increased protein levels with no apparent contribution of transcription (Figure 3E).”; I fail to see the data for “no apparent contribution of transcription” in Fig 3E.

- Page 12, Line 377: “promoter proximal pausing of RNAPII (p.val= $2E-22$)”; Where is this analysis visualized or based on? I do not see any ChIP-Seq of Pol2, or any other analysis that support this statement or p-value.

- Page 13 Line 429: typo: “2iLikewise”.; Fig 7c (and maybe elsewhere): Marcks = Marks

- Figs 8a, 8b show the main gene regulatory events in 2i and Cdk8/19i, that as far as I understand might lead to all other events as shown in Fig 8c. However, Figs 8a and 8b are mainly depicting a small analysis as performed in Fig 7, while there are much more (gene-)regulatory events analyzed in the manuscript. I do not understand why the authors depict Figs 8a and 8b separately, while only focusing on ERF/SIC and SEC/MED. What is the relation of Figs 8a and 8b to Fig 8c? The authors might consider making one final model, which incorporates the three panels that are now separated.

Reviewer #2 (Remarks to the Author):

The manuscript by Martinez-Val et al describes the molecular characterization of mouse ES cells that are induced to transition to the naïve pluripotent state by treating cells in 'classical' 2i conditions, and by inhibiting CDK8/19. The latter condition was identified recently by the same authors as an alternative to induce naïve pluripotency. The present study aimed to identify and compare the molecular trajectories induced by each treatment, by performing proteomic, phosphoproteomic and metabolomics analyses each in a time-course fashion. The authors observe large similarities, but also distinct differences pointing to mechanistic differences how naïve pluripotency is achieved. Specifically, they observed that CDK8/19i acts on the transcriptional machinery to regulate expression of distinct genes, while 2i results in phosphorylation of downstream targets in pluripotency network.

This manuscript complements a recent study by the same authors that is forthcoming in Nat Cell Biol (judged from ref #4 in the manuscript, no acceptance letter was provided), where they identified that inhibition of CDK8/19 induced naïve pluripotency. That paper focused on the cellular and functional comparison of naïve cells produced by 2i and CDK8/19i, complemented with proteomic and phosphoproteome characterization of these cells. The current study significantly extends these molecular studies by performing in-depth proteomic, phosphoproteomic and metabolomics analyses in a time course manner, to study the trajectory of cells during their transition to the naïve state. As such, a number of studies have been performed in the past studying 2i-mediated induction of pluripotency by genomic and (phospho)proteomic analyses, however, as far as I am aware, this has not been done across the 3 molecular layers presented here (in addition to transcriptome data lifted from previous studies). In addition, the detailed comparison to CDK8/19i treatment adds significant value by pointing to interesting mechanistic properties in either condition. Overall the study is well designed, results are clearly described, and figures are exquisitely well-crafted. The paper is packed with details, which will be valuable for the stem cell and data modeling communities alike. The conclusions are compelling, and will likely lead to further detailed investigations in the field. Apart from some minor comments I recommend publication in the journal.

Comments:

1. The way the data are presented in the first results section, the authors steer towards a conclusion that the trajectories producing 2i and CDK8/19i cells are highly similar. This is a bit misleading since many differences exist – as is already apparent from Fig 1, and obviously as is demonstrated in detail in the remainder of the manuscript. Clearly it is a distinct perspective chosen by the authors, but this reviewer was set on the wrong path, seeing obvious differences e.g. in fig 1f which were largely ignored in the text.

2. Page 4 line 114: ‘The most up-regulated protein after 14 days of 2i, GABARAPL2, was also the highest change in CDK8/19i’: this is not clear from fig 2a.

3. Fig 3a: how do the authors explain the specific (static) behavior of the ECT1 complex, while other ECT complexes change dynamically?

4. It is unclear how many replicates were performed for each of the time course analyses – this should be clarified.

Reviewer #3 (Remarks to the Author):

The authors provide careful in depth analysis of new datasets characterizing embryonic stem cells transitioning to two chemically-induced states of naïve pluripotency. The thorough comparison of proteomic changes occurring in cells transitioning to 2i or CDK8/9i provide strong evidence that these two interventions induce similar proteome remodeling. While these datasets have the potential to provide interesting insight into the biology of the naïve pluripotent state—or to the mechanisms of action of these chemical perturbations—at present the manuscript does not contain the experiments required to support the stated conclusions.

Throughout the manuscript, the authors rely on correlative data based on proteomic/metabolic snapshots to draw functional conclusions. All of the conclusions that are drawn from correlative evidence but not tested (some examples of which are listed below) should be restated or tested experimentally. The discussion should also be modified accordingly.

a. The authors point to altered expression of lysosomal genes to support the claim that proteome remodeling by autophagic flux is a feature of naïve pluripotency. However, autophagic flux is not assessed and the contribution of autophagy to proteome remodeling is untested. Similarly, whether there is indeed activation of a proteostasis axis or degradation of nuclear proteins is not tested.

b. The authors also suggest that degradation of LIN28 releases mRNAs from repressive control and thereby contributes to many of the protein changes seen, but again, the role of LIN28 is not tested. The authors also make several conclusions about translation rate across conditions, but translation is

not measured. A recent paper from the Stunnenberg group published in Nature Communications performed detailed analysis of translation and transcription in S/L vs 2i-cultured ESCs. These data argue that the majority of proteome changes are driven by transcription, findings that do not appear to support the conclusions of the present manuscript.

c. Steady state metabolite levels, even in combination with protein expression, cannot be used to infer active metabolic fluxes. Therefore, the conclusions that ESCs use choline to generate 1C units or that ESCs enhance respiration and beta oxidation/diminish glycolysis remain premature.

d. The authors show that ERF targets are de-regulated, but whether ERF is directly involved in these changes is not tested, limiting the ability to conclude that ERF is an early downstream effector of 2i. I also found it a bit difficult in the ERF section to distinguish which conclusions were drawn from the dataset and which were already reported in the literature (e.g. ERF favoring expression of targets of pluripotency signaling, co-activating ESRRB, repressing negative regulators, etc...)

Minor

- The methods note that missing values were imputed in metabolomics data. To my knowledge, this is not standard in the field. Out of an abundance of caution for future users of the data, could the authors note which values were imputed in the supplemental table?

- The authors should modify the text on p. 2 line 37 which suggests that “metastable serum ESCs” represent a primed, post-implantation epiblast-like state. As shown in ref. 3, these cells contain a mix of naïve and more primed cells.

- Supplementary Fig. 1E and S5D: the z-score axis is confusing and might be swapped? Is a negative z-score a protein that goes up? If so, this should be clarified, especially as Fig. 1d appears to have positive z-scores more intuitively represent positive fold changes.

- All figures should have a brief key specifying the red/blue scheme so that readers can quickly identify each condition.

- Fig. 3C: is color missing for the right four samples?

- Is it known that ERK regulates ERF stabilization and nuclear translocation? If so, the appropriate reference should be cited on p.11. If not, this statement should be tested experimentally.

- There appears to be missing text on p. 13 line 429.

REVIEWER COMMENTS

Reviewer #1 (Remarks to the Author):

Within the manuscript “Dissection of two routes to naïve pluripotency using different kinase inhibitors” by Martinez-Val et al., the authors compare two strategies to induce naïve pluripotency in mouse ESCs: the canonical “2i” method, and a novel method using a Cdk8/19 inhibitor as very recently pioneered by the authors themselves (soon to be published in NCB). In particular, the authors use profiling of the (phospho)proteome and metabolome during transition towards naive pluripotency to characterize potential regulatory mechanisms and key regulatory events associated with naïve pluripotency. I much enjoyed reading the manuscript, it is very well written with highly informative figures, clear in-depth analysis and a very straight storyline. Also, it is immediately clear that the authors are authorities on the subject, as they manage to provide a very comprehensive view on the event leading to naive pluripotency, including (to me partly unexpected) roles for molecules such as mTOR, LIN28, CIC, ERF etc. Also, they combine a large range of previous observation into one holistic view. I noticed that the manuscript exclusively leans on (analysis of) the (phospho)proteome and metabolome profiling, and therefore this manuscript might also label as a “data resource” paper. However, this directly shows two minor weakness of the current paper: (i) it is highly specialized in a direct, rather descriptive comparison of two methods to induce naïve pluripotency in mESCs: are there any general principles from such extensive profiling to be conveyed towards a more general audience; (ii) none of the findings have been validated or (experimentally) followed-up. Any attempt to address these two issues would significantly improve the manuscript (in particular independent validation of the key findings).

We thank the reviewer for their supportive comments on our manuscript. Indeed, our work is fundamentally based on high-throughput mass spectrometry data and it is presented as a description of molecular events involved in this biological process. As pointed out by this reviewer, some of these findings are novel and represent a valuable resource for the stem cell community. We agree that some of the events and processes described here might apply to other contexts. Indeed, in our preceding publication by Lynch et al Nat Cell Biol (PMID: 32989249), we showed that the CDK8/19i also shifts human ESCs to the naive state, so we expect that some of our findings might also be present in this system. We also believe that our results might be valuable to a broader audience and to illustrate our point, we have included a paragraph in the discussion where we speculate about the translation of some of our findings to the role of kinase regulation and cancer (see lines 530-549).

We also agree on the importance of independent experimental validation of our MS-based findings. Given the major changes found for mitochondrial proteins in response to 2i and CDK8/19i, we have now examined mitochondria morphology (TEM) and biomass (Mito-tracker) and did not observe differences with respect to S/L (see New Suppl Figure S3A-B), in agreement with the presence of post-transcriptional regulation found in our data for this organelle. Our data also suggested a major up-regulation of all five ETC complexes in 2i which might support an increase in OXPHOS metabolism. Interestingly, CDK8/19i failed to up-regulate Complex I (see comment 3 from Reviewer 2) (see Figure 3A), which is the entry point of the redox process and generates up to 40% of the ATP. Therefore, we decided to investigate this difference further by performing a specific assay of ATP content per cell, in order to assess the specific differences predicted by our proteomic analysis for ETC Complex I in mESCs in S/L, 2i or CDK8/19i (New Figure 3B). We found that 2i cells were more sensitive to rotenone, a Complex I inhibitor, than S/L cells (New Figure 3B). CDK8/19i, on the other hand, were resistant to this drug. In agreement with our proteomic data, these results confirm the different

dependency on Complex I between 2i and CDK8/19i for ATP production. Furthermore, we have analyzed the subcellular localization of ERF. Using immunofluorescence, we found that inhibition of ERK in 2i leads to its rapid nuclear stabilization (New Figure 7B). Consistent with our proteomic data predictions, the new immunofluorescence confirms that CDK8/19i has no effects on ERF. Finally, we have confirmed the increased in autophagy levels in both 2i and CDK8/19i, by assessing LC3B-II cleavage by western blot (New Suppl. Figure S2B).

Other major issues:

1) The authors compare two strategies to induce naïve pluripotency, but they never show clear validation of their timecourse using markers that have been extensively published for 2i and serum mESCs (and Cdk8/19i mESCs?). In particular, at the start of their manuscript, it would be highly informative (/essential) to show 5-mC and 5-hmC levels of the serum mESCs as being used, as well as after 14 days culturing in either 2i and Cdk8/19i, so to validate the system in their hands (at least in 2i this should show a very clear drop). This is also important as the authors later refer to 5-mC, but they never assayed it in their own system. Also, the authors should show a direct comparison of the dynamic changes of a few naïve and serum-specific markers between their study and published data to validate their system from the start on (rather than focusing on their own data only using z-scores).

We fully agree on the importance of validating our cellular system, especially, given the novel role of CDK8/19i in the context of pluripotency. In the original submission, in the first paragraph of the results section, we showed a direct comparison between our data and other published data sets of mESCs in 2i (PMID22541430, PMID26431182, PMID27345836, PMID27345836, PMID2655505): *“We mapped transcriptomic signatures of 2i-treated cells in our temporal-resolved proteome data and found that RNA changes showed overall concomitant protein alterations already after 48 h of 2i treatment, which were also recapitulated in CDK8/19i”*. In addition, we performed a similar comparison with gene signatures from specific stages of mouse development from Boroviak et al (PMID:26555056): *“Likewise, post-implantation genes correlated with repressed proteins in both 2i and CDK8/19i, whereas pre-implantation genes (E3.5 and E4.5) showed positive trends”*. The results from all these comparisons were shown in Suppl. Figure 1D and demonstrated, as expected, a good agreement between our system and published data. To make these comparisons more evident, we have now improved Suppl Fig 1D and included a new panel (Suppl Fig 1E) where we highlight the changes of specific genes markers in our MS data and in several other publications.

Regarding the DNA methylation levels. In our preceding publication by Lynch et al. Nat Cell Biol 2020 (PMID: 32989249), we analyzed the levels of 5-mC in mESCs grown in S/L, 2i and CDK8/19i culture media. We showed that 2i but not CDK8/19i significantly decreased global DNA methylation levels (Figure 5A in Lynch *et al.*). These analyses were performed in exactly the same four mES cell lines that we MS-profiled in our manuscript by Martinez-Val *et al.*

[Redacted]

2) I was very surprised to read that the authors used the 2i and Cdk8/19i on top of the serum, rather than replacing the serum. Notably, the general accepted and widely used naive pluripotent state represent “2i” conditions without serum. In the conditions used in the paper, the mESCs are in a sort of semi-naïve pluripotent state (which is why it is important to extensively validate this system, see point 1). It would be informative if the authors provide a rationale in the paper on why they preferred this setup by adding the inhibitors on top of the serum. More importantly, I feel it is important that the authors validate their main findings in serum-free cultures. It would be highly interesting if the authors find some additional differences.

We are glad that this reviewer points out the relevance of serum for maintaining mESCs in the naive state. The undefined composition of serum results in cell metastability because conflicting signaling pathways are activated. 2i alone is sufficient to drive many of the effects typical of the naive state. Consistent with this, while it is a common practice to remove serum from media and expand mESCs in 2i/LIF-based formula, a number of reports on naïve pluripotency do not do this (for example Finley et al 2018 Nat Cell Biol). In our case, we have a particular purpose for using serum in our culture media. We preferred to keep serum in all our experiments in order to focus on analyzing the direct effects of CDK8/19i on naive pluripotency. Therefore, the specific effects of each kinase inhibitor could be studied in isolation, without any putative contribution of serum-free conditions on the cells. Nevertheless, in our prior report by Lynch *et al* Nat Cell Biol 2020 (PMID: 32989249), we have comprehensively compared serum+ versus serum-conditions, showing that the typical morphology of mESCs and the homogenous expression of Nanog-GFP in 2i+LIF (serum-free, KSR) was recapitulated also in CDK8/19i+LIF (serum-free, KSR) (Fig.1A and Extended Data Fig.1A in Lynch et al). We also showed that the transcriptome of CDK8/19i overlaps significantly with 2i conditions, in both serum-containing and serum-free media (Fig.4A, Extended Data Fig.3E,F and Supplementary Table 2 in Lynch et al.). To better address this point, we have now re-analysed the above-mentioned data in more detail and found that global transcriptional changes induced by CDK8/19i in the presence of serum are phenocopied when CDK8/19i was supplemented in a serum-free based media (KSR) (Pearson=0.607). This result is very similar to the effects of 2i in the presence or absence of serum (Pearson=0.543). Indeed, with respect to reference serum/LIF conditions, CDK8/19i and 2i show a much better correlation in KSR-based media (Pearson=0.714) than in serum-containing media (Pearson=0.337), demonstrating that serum deprivation has an effect on its own and, consequently, to analyze the effects of CDK8/19i in naive pluripotency, our experiments are better performed in the presence of serum. Moreover, the major findings described in our work are also reproduced in serum-free conditions. For instance, we found that Lin28b is down-regulated in response in both 2i and CDK8/19i with and without serum. Accordingly, we have seen that mitochondrial proteins are more up-regulated at the protein level

than the mRNA level also in serum-free conditions (see the analysis of Atlasi's data below), confirming that post-transcriptional regulation is a bona fide feature of the naive pluripotent state independent of serum. We thank this reviewer for bringing this up and, in the revised manuscript, we have explained the rationale behind our cell culture media choice (first paragraph Results section).

3) The analysis and representation of the phosphoproteomics is unclear to me. As far as I could find, the authors analyze phosphopeptides. But how do the authors discriminate between downregulation of a total protein versus specific changes in phospho-sites within that same protein. For example (but relevant for all such analysis): Are the phospho-sites of MTOR in Fig 6d specifically absent/downregulated in naïve pluripotency, or is simply the whole MTOR protein downregulated? If the authors did not account for total protein changes, they should. Also, in that case, it would be preferred to show the proteome changes in parallel, or at least the fold-change in total protein after 1d of 2i or Cdk8/19i for which the authors have the data.

We are aware of the importance of calibrating phosphorylation changes by protein levels. Although we cannot completely rule-out the contribution of changes in protein abundance in the measured p-sites, there are two major points which suggest that this important issue is minimized in our data:

(i) Phosphorylation changes were sampled at the very early moments of adding the 2i or CDK8/19i (0.5, 1, 2 and 6 hours). This should enable us to capture the major signal transduction events driven by phosphorylation before gene expression changes take place. In support of this hypothesis, Yang et al (PMID: 31078527) performed a multi-omic profiling of naive mESCs exiting pluripotency and showed that widespread proteome changes were only evident after 12h. They demonstrated that phosphorylation cascades precede ordered waves of epigenomic, transcriptomic, and proteomic changes (Fig 2D in Yang *et al.*). Although this was found in the reverse process (the transition of naive towards “primed EpiSC”), a similar behavior of molecular events could be expected in our system (“primed S/L” to naive). As suggested by this reviewer, we have compared overall phosphorylation changes with protein levels at day 1 and did not find any correlation (figure below, panel A). We found similar results in Yang's data for matched phosphorylation-proteome data at 30 and 60 min (figure below, panel B). In agreement with Yang et al. (Fig 2E), we also found that, in both 2i and CDK8/19i, the magnitude of

phosphorylation changes in the first 6h is superior to that of protein changes at day 1 (panels C-D). Moreover, proteomic changes at day 1 are significantly minor and notably smaller compared to subsequent time points (2, 4, 7, 10 and 14 days), suggesting that in the first 6h of 2i and CDK8/19i, regulatory events are mainly driven by changes in phosphorylation rather than protein abundance.

(ii) Molecular information can often be transmitted in a site-specific manner: different sites in the same protein have different functions and, therefore, can be subject to differential regulation. Consequently, if changes in protein abundance are the major determinants in the regulation of its function (at a given time/stimuli), one might expect that all the identified phosphopeptides for this protein should exhibit nearly identical kinetics. However, we found in our data a very large number of proteins that showed site-specific regulation of their p-sites (see pie charts, below). Nevertheless, functional cooperativity within proximal phosphorylation sites is also a prevalent regulatory mechanism of proteins. This can also explain the nearly identical kinetics observed for certain p-sites, (S1621 and S1622 in mTOR. Fig 6D).

4) A recent manuscript by Atlasi et al. (referred to by the authors and also published in Nature Comm; PMID: 32238817) extensively mapped translational events in serum and 2i ESCs using ribosome profiling, which is very complementary to the current manuscript. Since part of the current manuscript also clearly focuses on the translational machinery (with various hypotheses and findings), the authors could very nicely incorporate the findings of Atlasi et al. to validate their findings (for example increased translation upon 2i or Cdk8/19i) or provide further evidence for some of their hypothesis regarding specific groups of proteins that are regulated at the translational level.

This was an excellent suggestion, and we have performed a comprehensive side-by-side comparison of our data with Atlasi et al. In their report, Atlasi and colleagues focused on the translational landscape of naïve pluripotency. They show that 2i possesses a higher ribosome density on a set of mRNAs but, surprisingly, this is not reflected in concomitant protein changes, which is attributed by the authors to translational buffering. In addition, they show that proteome changes between pluripotent states are mainly driven by transcriptional rewiring. In light of one of our major findings regarding the presence of post-transcriptional regulation in the naive state, Atlasi's results seem somehow discrepant with our findings. We have now examined the work from Atlasi et al. and found strong evidence of post-transcriptional regulation in their data too, corroborating thereby our results and providing a potential mechanism which integrates the key findings of both studies.

In the analyses of RNA, Ribosomal Foot-Prints (RFP) and proteins levels, Atlasi attributed protein changes as “transcription, translation or post-translation” on the basis of defined cut-offs (Fig. 3D). The established cut-offs for “protein changes” were notably stricter (Fold-Change > 3) than those used for transcriptional (RNA-seq) and translational (RFP) changes (Fold-Change > 2). These different criteria might impose a bias that favored transcription as the main driver in defining the expression programs of pluripotent states. However, to better assess the contribution of each regulatory layer in gene expression control, we believe that smaller divergences between mRNA, RFP and protein levels should be accounted for. Indeed, using a 2-D enrichment approach that omits the need of arbitrary cut-offs, we have found notable differences between mRNA and RFP levels in Atlasi's data. This analysis reveals that genes regulated mainly at the translational level (RNA < RFP) are significantly enriched in endoplasmic reticulum and plasma membrane (FDR=0) (figure below, A-B). Importantly, these changes were only apparent after day 7, coincident with the down-regulation of Lin28 in Atlasi's time series (figure below, panel C). Therefore, these results support our findings, and others (PMID: 23102813), on the role of LIN28 as a suppressor of ER-associated translation in pluripotency.

B

T: GO_Term	Day 1 (2i vs S/L)				Day 3 (2i vs S/L)				Day 7 (2i vs S/L)				Long-Term (2i vs S/L)			
	RNA	RFP	P.val	FDR	RNA	RFP	P.val	FDR	RNA	RFP	P.val	FDR	RNA	RFP	P.val	FDR
integral to membrane	-0.508	-0.502	0	0	-0.313	-0.128	0	0	-0.275	0.053	0	0	-0.069	0.100	0	0
intrinsic to membrane	-0.511	-0.505	0	0	-0.318	-0.129	0	0	-0.274	0.058	0	0	-0.070	0.107	0	0
endoplasmic reticulum part	-0.475	-0.448	0	0	-0.308	-0.096	0	0	-0.292	-0.094	0	0	-0.086	0.111	0	0
membrane part	-0.405	-0.394	0	0	-0.249	-0.052	0	0	-0.216	-0.043	0	0	-0.051	0.087	0	0
ribonucleoprotein complex	0.159	0.205	2.20E-16	2.44E-14	0.190	0.144	6.51E-07	3.28E-05	0.048	-0.209	0	0	-0.205	-0.085	6.04E-11	1.34E-08
nucleobase-containing compound metabolic process	0.125	0.140	6.94E-12	1.48E-08	0.096	0.022	4.29E-09	5.34E-06	0.037	-0.118	0	0	-0.028	-0.056	4.41E-10	1.94E-06
membrane	-0.190	-0.191	0	0	-0.124	-0.030	1.07E-14	1.97E-12	-0.088	0.073	0	0	0.013	0.071	2.44E-07	1.81E-05
cellular nitrogen compound metabolic process	0.116	0.132	3.10E-11	5.44E-08	0.100	0.022	3.44E-10	6.04E-07	0.039	-0.111	0	0	-0.020	-0.063	8.89E-09	2.01E-05
RNA binding	0.090	0.191	2.20E-10	6.77E-08	NaN	NaN	NaN	NaN	-0.022	-0.253	0	0	-0.177	-0.167	5.48E-08	2.09E-05

Noteworthy, the 2D-enrichment analysis of Atlasi's data did not show any differences in the translation of mitochondrial proteins. However, when comparing protein vs RNA levels, we found a significant difference, as mitochondrial components were more up-regulated at the protein level than they were at the mRNA level ($p.val=2.8e-19$) (figure below, panel D). As expected, this divergence is synchronized with the down-regulation of LIN28, validating again our results. In addition, we obtained very similar results when comparing RFP and protein levels (figure below, panel D). Therefore, Atlasi's data independently confirm our findings that post-transcriptional regulation of mitochondrial proteins, under control of RNAbp's such as LIN28, is a major feature of naïve pluripotency.

An interesting question arises from these results concerning the true contribution of translation (as measured by RFP) to the final protein concentration levels (as measured by MS). Given that RFP is a direct reflection of translational activity, it is intriguing that the up-regulation of mitochondrial protein levels (MS) is not detected in the RFP of Atlasi and colleagues. Below, we propose several explanations (or a combination of them) which might explain this result:

- The naïve state may increase the stability of mitochondrial proteins (for instance, by means of post-translational modifications). However, the apparent role of LIN28 degradation in the up-regulation of these proteins argues against this hypothesis.

- Under specific circumstances such as stress, cells undergo global changes in transcription or translation rates. Accounting incorrectly for these changes can distort gene-specific measurements. In this scenario, pulsed-SILAC is reported to detect changes in protein synthesis more accurately than RFP (PMID: 28578850). Given that 2i exhibits lower transcription and translation rates than S/L (PMID: 27880763), a result confirmed in our data (Fig 3F), it may be conceivable that normalization of RFP in Atlasi's data might lead to inaccurate estimates of translation efficiencies as seen before by others (PMID: 28662383) and might mask actual changes in the footprints of mitochondrial proteins.
- RFP is normally measured in bulk cell populations, providing an average estimate of translational activity. However, translation is often spatially regulated in organelles (PMID: 31230715). In fact, proximity-specific ribosome profiling has revealed pervasive co-translational targeting to ER (PMID: 25378630) and mitochondria (PMID: 25378625). Therefore, a local increase in translation at the surface of these organelles might be unnoticed when using a whole cell RFP strategy like in Atlasi.
- Translationally inactive mRNAs can be engaged into ribonucleoproteins and stalled/paused in polysomes due to physiological surveillance, stress and regulatory mechanisms (PMID: 29310120). While inactive ribosomes unbound to transcripts do not present a problem, Riboseq does not differentiate actual protected footprints of translating polysomes from RNA fragments sequestered in inactive/stalled ribosomes, leading to possible misinterpretations of translation occupancies (PMID: 30355487). In S/L mESCs, ribosomes in ER and mitochondria are pre-loaded with their target mRNAs but, because of LIN28 binding, do not undergo active translation. Since the RFP approach employed by Atlasi cannot distinguish inactive from active ribosomes, the overall net change for mitochondrial mRNAs footprints upon LIN28 degradation in 2i might therefore remain constant. Indeed, translational control of pre-existing mRNAs allows faster protein synthesis than the transcription of new mRNAs, which may explain the prominence of translational regulation in stress responses (PMID: 20965418 and PMID: 27015305).

We agree with reviewers 1 and 3 that this is an important and interesting point for discussion in our manuscript and we thank them for bringing this up. In the revised version, we discuss now the results from Atlasi and colleagues in the context of the post-transcriptional and translational regulation in the naïve state (see lines: 476-499) and present the results in new Suppl Figure S10.

5) On page 11, line 357: "Hence, our data suggest that 2i might signal directly to EZH2 to regulate PRC2-independent processes." However, there is clear evidence that EZH2 is also directly involved in maintaining naïve pluripotency as part of the PRC2 complex (PMID: 30472157). The authors might want to take this study into consideration regarding this specific issue.

The reviewer is right to remark the role of EZH2, as part of the PRC2 complex, in the maintenance of the naïve state by protecting 2i-ESCs from primed-like features like DNA methylation. In fact, our data agrees with some of the findings in PMID:30472157. We found that 2i, but not CDK8/19i up-regulates EZH1 and EZH2, whereas JARID2 is prominently down-regulated (same changes were found in PMID:30472157). Although our measurements are based on total protein levels, unlike the chromatin-bound fraction analyses performed in PMID:30472157, the differences in PRC2 composition between 2i and CDK8/19i could explain/contribute to the differences in DNA methylation reported in our work between 2i and CDK8/19i. However, we would like to clarify that our statement on page 11 suggests an additional role of EZH2 out of the PRC2 complex, without neglecting its nuclear role in H3K27me3 deposition and the naïve state. Following this reviewer's suggestion, we have now re-phrased our

statement to accommodate both findings and appended the reference PMID: 30472157 (which in our original submission was only cited in the introduction).

Minor issues:

- For Fig 1F and similar figures further on, the authors should include within the figure the number of proteins included in each cluster.

We appreciate the suggestion and we have now included the number of proteins in each cluster and in all relevant figures throughout the manuscript.

- Page 3, line 77: A p.val of 2.7E-20 seems very highly significant to me. On the other hand, I tend to agree that an overlap of 12 proteins out of ~1100 proteins does not seem significant, but this is not supported by such p-value?

The p-value from the hypergeometric test used here evaluates the significance of the overlap between two gene lists. The test however determines if the overlap is higher or lower than expected (over-representation and under-representation respectively). This Representation-Factor (RF) is indicated in our figures together with their associated p-values (“Supplementary Figure S1C”). The large overlap (RF>1 and low p-values) found between up-regulated proteins in both 2i and CDK8/19i (and also between down-regulated proteins) indicated that both treatments induce similar changes. On the other hand, when we examined proteins showing changes in opposite directions (up-regulated in 2i but down-regulated in CDK8/19i and vice versa), we found no overlap at all (RF<1 and low p-values) (*i.e.* less proteins than expected by chance). Therefore, this second comparison further demonstrates the large similarities in the changes induced by both treatments.

- Page 6, line 182: “several targets of LIN28 showed increased protein levels with no apparent contribution of transcription (Figure 3E).”; I fail to see the data for “no apparent contribution of transcription” in Fig 3E.

We agree that this sentence is unclear. We have now re-phrased the text to: “*Indeed, we found several proteins regulated in a post-transcriptional manner that showed opposite kinetics to LIN28 in response to both treatments. Consistent with this, several mitochondrial proteins mirrored the down-regulation of LIN28 in a significant manner*” (Figure 3F and Supplementary Figure 3F)” (see lines:206-208). To clarify this further, we have now included a new supplementary figure S3D showing the temporal expression profiles and number of proteins that, according to our data, are potentially regulated in a post-transcriptional manner, highlighting those that are mitochondrial or have been defined as LIN28 targets.

- Page 12, Line 377: “promoter proximal pausing of RNAPII (p.val=2E-22)”; Where is this analysis visualized or based on? I do not see any ChIP-Seq of Pol2, or any other analysis that support this statement or p-value.

This analysis shows the significance of the overlap between the targets found to be regulated by promoter proximal pausing of RNAPII in Williams et al (PMID25773599) and targets repressed by ERF in Mayor-Ruiz et al (PMID29650524). We have now shown these results in Suppl. Figure 9C and explain it accordingly in the figure legend.

- Page 13 Line 429: typo: “2iLikewise”.; Fig 7c (and maybe elsewhere): Marcks = Marks

This has now been corrected.

- Figs 8a, 8b show the main gene regulatory events in 2i and Cdk8/19i, that as far as I understand might lead to all other events as shown in Fig 8c. However, Figs 8a and 8b are mainly depicting a small analysis as performed in Fig 7, while there are much more (gene-)regulatory events analyzed in the manuscript. I do not understand why the authors depict Figs 8a and 8b separately, while only focusing on ERF/SIC and SEC/MED. What is the relation of Figs 8a and 8b to Fig 8c? The authors might consider making one final model, which incorporates the three panels that are now separated.

We thank the reviewer for this suggestion. Following this advice, we have now merged all panels in a single model and included some of the other gene regulatory findings that are mentioned in the main text.

Reviewer #2 (Remarks to the Author):

The manuscript by Martinez-Val et al describes the molecular characterization of mouse ES cells that are induced to transition to the naïve pluripotent state by treating cells in ‘classical’ 2i conditions, and by inhibiting CDK8/19. The latter condition was identified recently by the same authors as an alternative to induce naïve pluripotency. The present study aimed to identify and compare the molecular trajectories induced by each treatment, by performing proteomic, phosphoproteomic and metabolomics analyses each in a time-course fashion. The authors observe large similarities, but also distinct differences pointing to mechanistic differences how naïve pluripotency is achieved. Specifically, they observed that CDK8/19i acts on the transcriptional machinery to regulate expression of distinct genes, while 2i results in phosphorylation of downstream targets in pluripotency network.

This manuscript complements a recent study by the same authors that is forthcoming in Nat Cell Biol (judged from ref #4 in the manuscript, no acceptance letter was provided), where they identified that inhibition of CDK8/19 induced naïve pluripotency. That paper focused on the cellular and functional comparison of naïve cells produced by 2i and CDK8/19i, complemented with proteomic and phosphoproteome characterization of these cells. The current study significantly extends these molecular studies by performing in-depth proteomic, phosphoproteomic and metabolomics analyses in a time course manner, to study the trajectory of cells during their transition to the naïve state. As such, a number of studies have been performed in the past studying 2i-mediated induction of pluripotency by genomic and (phospho)proteomic analyses, however, as far as I am aware, this has not been done across the 3 molecular layers presented here (in addition to transcriptome data lifted from previous studies). In addition, the detailed comparison to CDK8/19i treatment adds significant value by pointing to interesting mechanistic properties in either condition. Overall the study is well designed, results are clearly described, and figures are exquisitely well-crafted. The paper is packed with details, which will be valuable for the stem cell and data modeling communities alike. The conclusions are compelling, and will likely lead to further detailed investigations in the field. Apart from some minor comments I recommend publication in the journal.

We really appreciate this reviewer’s recommendation to publish our work in Nature Communications. Certainly, the current manuscript complements and extends some of the findings described in Lynch et al Nat Cell Biol (now published PMID: 32989249). In the original submission, we did not include an acceptance letter, but attached instead the final version of the manuscript as suppl. material. We apologize for this inconvenience.

Comments:

1. The way the data are presented in the first results section, the authors steer towards a conclusion that the trajectories producing 2i and CDK8/19i cells are highly similar. This is a bit misleading since many differences exist – as is already apparent from Fig 1, and obviously as is demonstrated in detail in the remainder of the manuscript. Clearly it is a distinct perspective chosen by the authors, but this reviewer was set on the wrong path, seeing obvious differences e.g. in fig 1f which were largely ignored in the text.

Certainly, given the common final outcome of both methods (*i.e.* stabilization of mESCs in the naive state), we believe that common changes to both 2i and CDK8/19i are important because they are conserved in two different approaches. Likewise, because each drug targets different pathways (extracellular signaling in 2i vs. enhancer stimulation in CDK8/19i), some differences should be expected. As noted by the reviewer, these differences are already evident in Fig. 1F and we briefly

discussed them in the text (GO enrichment results and Fig 1G). Later, throughout our manuscript, we focused on the differences that we consider most relevant for this process, such as DNA methylation. In an attempt to improve the readability of our manuscript, we have now changed some of the text in the first section of the Results which we think introduce the data in a more neutral and impartial mode. We thank the reviewer for this comment.

2. Page 4 line 114: ‘The most up-regulated protein after 14 days of 2i, GABARAPL2, was also the highest change in CDK8/19i’: this is not clear from fig 2a.

To clarify our point, we have now referenced this finding to the Supplementary Table S1A, where all the identified proteins can be ranked by their log₂FC (vs S/L) in each specific time-point. GABARAPL2 is the most up-regulated protein at 14 days in both treatments.

3. Fig 3a: how do the authors explain the specific (static) behavior of the ECT1 complex, while other ECT complexes change dynamically?

Differential regulation of ETCs could be attributed to multiple factors. (i) mRNA levels of Complex I subunits did not show prominent differences between 2i and CDK8/19i, therefore we discard transcriptional mechanisms. (ii) Although other RNA binding proteins might be involved, the similar degradation of LIN28A in both inhibitors also excludes a differential post-transcriptional control. (iii) Proteins belonging to the mitochondrial import machinery are similarly regulated in both conditions, thus CDK8/19i-dependent defects in complex I protein import seem equally unlikely. (iiii) Complex I assembly is an intricate process controlled by at least 13 proteins (PMID: 27040506). Our data shows for instance that ACAD9 is similarly up-regulated in response to both treatments. However, NDUFAF7, an arginine methyltransferase that is essential in early development (PMID: 24838397), was only down-regulated in CDK8/19i. Therefore, we speculate that a differential regulation of Complex I assembly factors could lead to defects in its assembly process in CDK8/19i. In support of this hypothesis, NDUFAF1 (CIA30), an important Complex I assembly factor, was detected up-regulated only in CDK8/19i which could indicate an ongoing compensatory effort (PMID: 29285300). A brief explanation to this result has now been included in the revised version (see lines: 180-183).

In addition, as explained above to Reviewer 1, we have validated with additional experimentation the functional consequences of this difference in Complex I (New Figure 3B). Indeed, our results show now the different susceptibility of 2i and CDK8/19i to inhibition of Complex I, suggesting a differential dependency on Complex I for ATP production between these cells.

4. It is unclear how many replicates were performed for each of the time course analyses – this should be clarified.

Instead of measuring the same cell line several times (*i.e.* biological replicates), we chose to analyse four different mouse embryonic stem cell lines (namely ZS, TNGA, TON and BL.6). Given that each of these cell lines possess different genetic backgrounds, we believe that significant proteomic changes identified using this experimental set up are therefore more robust than multiple analyses of just a single cell-line. The use of four different cell lines was mentioned several times throughout the manuscript, but to better clarify our choice of experimental set up, we have now included an explanation in the Methods section of the revised manuscript (lines: 832-833).

Reviewer #3 (Remarks to the Author):

The authors provide careful in depth analysis of new datasets characterizing embryonic stem cells transitioning to two chemically-induced states of naïve pluripotency. The thorough comparison of proteomic changes occurring in cells transitioning to 2i or CDK8/9i provide strong evidence that these two interventions induce similar proteome remodeling. While these datasets have the potential to provide interesting insight into the biology of the naïve pluripotent state—or to the mechanisms of action of these chemical perturbations—at present the manuscript does not contain the experiments required to support the stated conclusions. Throughout the manuscript, the authors rely on correlative data based on proteomic/metabolic snapshots to draw functional conclusions. All of the conclusions that are drawn from correlative evidence but not tested (some examples of which are listed below) should be restated or tested experimentally. The discussion should also be modified accordingly.

We are grateful that this reviewer sees the potential value of our data to improve our understanding of the naïve pluripotent state. This is one of the largest MS-based characterizations of the mechanisms involved in naïve pluripotency, and the resource-type of work that is reported here provides novel insights for other laboratories working in this field. For instance, the description of early downstream phosphorylation events in response to these kinase inhibitors is novel and some of these targets could be followed-up by others. Yet, we agree with the Reviewer and we fully acknowledge that further experimental validation of our MS-based findings is important. To directly address this point in the revised version, we have performed a number of additional experiments and re-analyzed published data that confirm and extend four of the major findings derived from our proteomic analysis regarding the downstream transcriptional response of 2i, the differences in the metabolic state of 2i and CDK8/19i cells, the increase autophagy levels in naïve cells and the presence of post-transcriptional regulation in mitochondrial proteins.

Because of their key role in our work and the suggestion of Reviewer 2, see above, we examined mitochondrial proteins in the data sets from Atlasi *et al*, and found that they are also subject to significant post-transcriptional regulation (see Comment 4 to Reviewer 2) (New Supplementary Figure S10 in the revised version), validating therefore our results. Then, using mito-tracker and electron microscopy, we have now observed that mitochondrial biomass and morphology in both 2i and CDK8/19i is similar to S/L mESCs (New Supplementary Figure S3A-B), authenticating that the up-regulation of mitochondrial proteins responded mainly to actual changes in the protein content rather than an increase of the mitochondrial biomass. In addition, we followed-up on one of the main differences between 2i and CDK8/19i with respect to the mitochondria, the differential expression of Complex I (Figure 3A). We have now performed an assay to measure ATP content per cell in mESCs in S/L treated with 2i or CDK8/19i, to which we added increasing concentrations of rotenone, a strong Complex I inhibitor (New Figure 3B). The CellTier-Glo® Luminescent assay determines ATP levels, which correlate with the percentage of metabolically active cells. From this assay, we observed that the ATP content rapidly and significantly decreased in 2i with increasing concentrations of rotenone, whereas CDK8/19i showed no effects on their ATP production (New Figure 3B). These results are in agreement with our proteomic results, and reflect the higher dependency on Complex I for ATP production in 2i. We have also confirmed the increased autophagy levels in our 2i and CDK8/19i systems by measuring the LC3 cleavage, which was more evident in the former. Lastly, we have confirmed by immunofluorescence that the inhibition of ERK in 2i results in the nuclear stabilization of ERF (New Figure 7B). In contrast, CDK8/19i had no effect on this downstream transcriptional effector (New Figure 7B) confirming another result from our proteomic data.

a. The authors point to altered expression of lysosomal genes to support the claim that proteome remodeling by autophagic flux is a feature of naïve pluripotency. However, autophagic flux is not assessed and the contribution of autophagy to proteome remodeling is untested. Similarly, whether there is indeed activation of a proteostasis axis or degradation of nuclear proteins is not tested.

We thank the reviewer for this comment. Our data (Fig. 2) shows that multiple proteins involved in lysosomal biogenesis and autophagy are up-regulated in both 2i and CDK8/19i cultures, albeit with noticeable differences between them (for instance, the V-ATPase subunits are only up-regulated in 2i). The relevance of autophagy for pluripotency has been studied before. For instance, Wu *et al.* Nat Cell Biol 2015 already showed that pluripotent mESCs have a higher basal level of autophagy than differentiated MEFs (Fig 1c,d in PMID: 25985393). Importantly, they show that mESCs in 2i/L have even higher autophagy levels than PRO cells maintained in S/L (Suppl Fig 1c,d in PMID: 25985393), which is in agreement with the increase of multiple autophagy and lysosomal proteins found in our current proteomic data.

Furthermore, we have now examined the expression levels of all these proteins in the data sets from Atlasi *et al* Nat Commun 2020 and found that many of them were also up-regulated throughout the time course of their experiment. These included important autophagy regulators (ULK1, LAMP2), known TFE3 targets (NPC2, CTSB, CTSD, LAMP1), multiple proteins from the CLEAR (Coordinated Lysosomal Expression and Regulation) network from TFEB (GBA, CTSA, PPT1, GLB1, PLBD2) and all the subunits from the V-ATPase, in great agreement with our proteomic data. However, the subunits from the major macro-molecular complexes involved in proteostasis barely changed their expression in Atlasi's data.

In addition, we have now assessed autophagy levels in our system by measuring the marker LC3-phosphatidylethanolamine conjugate (LC3B-II), which is recruited to autophagosomal membrane. In the western blot (new Suppl. Fig. S2B), we found the appearance of the LC3BII band in both 2i and CDK8/19i time course experiments indicating higher autophagy flux. We noticed however that this increase was more evident in 2i than in CDK8/19i, which might explain some of the proteomic differences found in autophagy components between both treatments and may suggest a higher autophagy flux in the former or an imperfect activation in the later. Overall, we feel that prior work on this matter (PMID: 25985393), the observation of remarkably similar changes to the ones found in our report in the data from Atlasi (PMID: 32238817) and the assessment of LC3 cleavage in our cell systems support our conclusions (see lines: 136-140).

b. The authors also suggest that degradation of LIN28 releases mRNAs from repressive control and thereby contributes to many of the protein changes seen, but again, the role of LIN28 is not tested. The authors also make several conclusions about translation rate across conditions, but translation is not measured. A recent paper from the Stunnenberg group published in Nature Communications performed detailed analysis of translation and transcription in S/L vis 2i-cultured ESCs. These data argue that the majority of proteome changes are driven by transcription, findings that do not appear to support the conclusions of the present manuscript.

We fully agree with the reviewer on the relevance of Atlasi's findings in the context of our work. In their report, Atlasi and colleagues show that 2i possesses a higher ribosome density on a set of mRNAs but, surprisingly, this is not reflected in concomitant protein changes, which is attributed by the authors to translational buffering. In addition, they show that proteome changes between pluripotent states are mainly driven by transcriptional rewiring. We have now re-examined the data from the Stunnenberg's lab and, in agreement with our current data, confirmed the presence of post-transcriptional regulation in their data too. A more detailed explanation and interpretation of these new analyses is given above in response to Reviewer #1's comment.

In Atlasi's report, the authors assigned protein changes as "transcription, translation or post-translation" using defined cut-offs (Fig. 3D). The established values for "protein changes" were notably higher (Fold-Change > 3) than those used for transcriptional (RNA-seq) and translational (RFP) changes (Fold-Change > 2). These different criteria might impose a bias that favored transcription as the main driver in defining the expression programs of pluripotent states. However, we believe that smaller divergences between mRNA, RFP and protein levels should be accounted for. Indeed, we have now re-analyzed Atlasi's data using a 2-D enrichment approach and found notable differences between mRNA and RFP levels. This strategy reveals that genes regulated mainly at the translational level (RNA < RFP) are enriched in endoplasmic reticulum and plasma membrane (FDR=0) (figure below, A-B). Importantly, these changes were only apparent after day 7, coincident with the down-regulation of Lin28 in Atlasi's time series (figure below, panel C). Therefore, these results support our findings, and others (PMID: 23102813), on the role of LIN28 as a suppressor of ER-associated translation in pluripotency. Remarkably, the 2D-enrichment analysis of Atlasi's data did not show any differences in the translation of mitochondrial proteins. However, when comparing protein vs RNA levels, we found that mitochondrial components were more up-regulated at the protein than the mRNA level (p.val=2.8e-19) (figure below, panel D). In support of our hypothesis, this divergence is synchronized with the down-regulation of LIN28. Therefore, Atlasi's data independently confirm our findings that post-transcriptional regulation of mitochondrial proteins, under control of RNAbp's such as LIN28, is a feature in naïve pluripotency. We thank Reviewers 1 and 3 for bringing up this important subject. In this revised version, we discuss these results in the text (see lines: 476-499) and in new Suppl Fig. S10.

c. Steady state metabolite levels, even in combination with protein expression, cannot be used to infer active metabolic fluxes. Therefore, the conclusions that ESCs use choline to generate 1C units or that ESCs enhance respiration and beta oxidation/diminish glycolysis remain premature.

To address the Reviewer's concern, we have toned-down our conclusions, and nuanced the text to reflect the point of the Reviewer. We recognize that our MS-based snapshots of metabolic and protein levels might not necessarily reflect the actual metabolite traffic present in this system. The main goal of our metabolomic analyses was, however, to complement our proteomic observations. Our results showed significant changes in the protein levels of numerous metabolic enzymes and thereby we sought to find out whether these had any impact on their metabolites concentrations. This strategy has been applied to study many other processes (PMID: 27669165, 28671190) and, indeed, we have confirmed previous results (*i.e.* glutamine utilization by mESCs), authenticating, to a certain extent, our approach. While using stable isotope tracers to study one-carbon fluxes would be highly interesting, this is a rather complex analysis that we feel is beyond the scope of this work. Though, in view of the large expression changes identified for some of the OCM-enzymes, this would be an exciting hypothesis to follow-up in future work. We agree that our data might certainly be premature to raise such conclusions, we have now modified the text to moderate our statements on the OCM.

d. The authors show that ERF targets are de-regulated, but whether ERF is directly involved in these changes is not tested, limiting the ability to conclude that ERF is an early downstream effector of 2i. I also found it a bit difficult in the ERF section to distinguish which conclusions were drawn from the dataset and which were already reported in the literature (e.g. ERF favoring expression of targets of pluripotency signaling, co-activating ESRRB, repressing negative regulators, etc...).

To directly address the Reviewer's concern, we have now clarified the related text and conclusions (see lines 387-390) as well as in Figure 7 legend. In this revised version, we have now confirmed the rapid re-localization and stabilization of ERF to the nuclei upon 2i addition to mESCs (New Figure 7B).

Importantly, these events are not observed in CDK8/19i, validating the functional relevance of ERF as downstream effector of the MEK-ERK pathway. We are sorry that the way that CIC/ERF data was presented in our original submission was not clear. The target genes for ERF and CIC were extracted from the ChIP-seq data sets performed in mESCs by:

- Mayor-Ruiz et al. (PMID: 29650524): ERF genomic binding sites were mapped by ChIP-seq in mESCs $RAS^{lox/lox}$ (-OHT vs +OHT). ERF-deficient mESCs were used as negative control. ERF peaks were associated with 5098 genes. Microarray analysis of mESCs $RAS^{lox/lox}$ (-OHT vs +OHT) were then used to map differentially expressed genes to ERF-bound genes and categorized as:
 - ERF-transcriptional repressor: 468 (out of the 5098 genes) were found to be down-regulated in RAS-deficient mESCs
 - ERF-transcriptional activator: 284 (out of the 5098 genes) were found to be up-regulated in RAS-deficient mESCs
- Weissmann et al. (PMID: 29844126):: CIC genomic binding sites were mapped by ChIP-seq in mESCs cultured in 2i/L media. CIC-KO mESCs were used as negative control. CIC peaks were associated with 112 genes.

To understand the gene expression changes due to stabilization of ERF and CIC in 2i, we mapped ERF and CIC target genes derived from the above publications in our proteomic and transcriptomic data sets. Our analyses confirmed some of the observations made by these authors regarding the repression of negative regulators of the MAPK pathway (Mayor-Ruiz et al and Weissmann et al.) and activation of pluripotency genes (Mayor-Ruiz et al). In addition, we expanded these findings by noticing that ERF and CIC repressed genes were also involved in transcriptional circuitries of the primed and formative pluripotent states.

Minor

- The methods note that missing values were imputed in metabolomics data. To my knowledge, this is not standard in the field. Out of an abundance of caution for future users of the data, could the authors note which values were imputed in the supplemental table?

We agree with the reviewer, so we have now indicated in the Supplementary Table S2 which values were imputed and how.

- The authors should modify the text on p. 2 line 37 which suggests that “metastable serum ESCs” represent a primed, post-implantation epiblast-like state. As shown in ref. 3, these cells contain a mix of naïve and more primed cells.

We thank the reviewer for this comment, and we have accordingly modified the text.

- Supplementary Fig. 1E and S5D: the z-score axis is confusing and might be swapped? Is a negative z-score a protein that goes up? If so, this should be clarified, especially as Fig. 1d appears to have positive z-scores more intuitively represent positive fold changes.

We thank the reviewer for noticing this out. Indeed, there was a typo in the legend (which was swapped). We have corrected it in the revised-version.

- All figures should have a brief key specifying the red/blue scheme so that readers can quickly identify each condition.

We agree with the reviewer's suggestion. We have now included the legend to those figures accordingly.

- Fig. 3C: is color missing for the right four samples?

The right four samples refer to mitochondrial proteins. They are displayed in black, following the same scheme color as in (new) Fig. 3C. However, we noticed that the corresponding samples are not indicated at the bottom of the chart. We have now changed the color code of panel 3C and 3D for better readability and consistency. We thank the reviewer for pointing this out.

- Is it known that ERK regulates ERF stabilization and nuclear translocation? If so, the appropriate reference should be cited on p.11. If not, this statement should be tested experimentally.

ERF subcellular location is known to be regulated due to the presence of growth factors that keep the protein inactive in the cytoplasm due to ERK-dependent phosphorylation. Upon ERK-inactivation (such as in 2i condition), ERF translocates to the nucleus to exert its function. This has been extensively described in literature: Sgouras et al. 1995; Le Gallic et al. 1999, and more recently in Mayor-Ruiz et al 2018. We have included all these references to clarify this issue. WE also have demonstrated experimentally this event in our system (New Figure 7D).

- There appears to be missing text on p. 13 line 429.

This has been corrected.

REVIEWER COMMENTS

Reviewer #1 (Remarks to the Author):

With respect to the paper “Dissection of two routes to naïve pluripotency using different kinase inhibitors” by Martinez-Val et al.”, I think the authors significantly improved their manuscript, it has become a very thorough and insightful manuscript.

There are however three final issues that should be addressed before publication:

- IMPORTANTLY: I forgot to mention in my previous review, but then and now the mass spectrometry proteomics data are not available in PRIDE using the reviewer login credentials, they seem to not have been submitted (“No Review Data”). All data (metadata AND processed data) should be submitted to a public database, and immediately made open access upon publication of this manuscript.
- In Fig 3a and Fig 3g (and elsewhere if not provided), the authors should add protein names to their heatmaps.
- I am very surprised that authors indicate that they much appreciate and value my comments, perform subsequent data analysis in the rebuttal to tackle the issue, but do not show these actual analyses/ figures anywhere in the paper (in particular if this happens multiple times). I feel the authors should seriously consider including the figures as shown in the rebuttal (1. Performed on my question “2) I was very surprised to read”; 2. Performed on my question “3) The analysis and representation of the phosphoproteomics is”; 3. Performed on the question of Reviewer 3 point a: “a. The authors point”). These are all highly useful analyses for readers that are likely having similar concerns as the reviewers have.

Reviewer #2 (Remarks to the Author):

The authors have addressed my concerns appropriately, and the new data have strengthened the manuscript. I therefore recommend acceptance for publication.

Reviewer #3 (Remarks to the Author):

In this revised manuscript, the authors add some new data, including measures of LC3 lipidation, ERF localization, and complex I dependency; include additional discussion, and temper the interpretation of the data, resulting in an improved submission. However, as pointed out by Reviewer 1 in the original submission, this paper remains a data resource paper without functional validation of any key findings. Whether any of the implicated pathways contribute to gene/protein regulation in ESCs is not tested. A prerequisite to publication is further modification of the text to ensure that no conclusions are drawn from correlative data. Once such critical modifications, detailed below, are included, this resource will be of interest to pluripotent stem cell enthusiasts.

Overall, the authors have done a good job modulating many of their conclusions to reflect their data. However, there remain many examples where the authors make conclusions based on correlative data. Some instances of drawing functional conclusions from descriptive data persist, and must be modified to reflect the actual data. Examples include:

- o P.5 the addition of LC3B-II levels is a welcome addition to demonstrate potential alterations in autophagy, but steady state levels of LC3 lipidation can reflect either changes in autophagosome synthesis or degradation and thus do not reflect flux (among other examples, see PMID: 25484088). The authors thus should revise their conclusions and avoid any implication that “increased autophagic flux is a common feature of naïve pluripotency.”

- o On p.5, the authors write, “our results reveal the rapid and concerted activation of a proteostasis axis induced under the early effects of 2i, leading to the degradation of nuclear proteins”. However, direct assessment of changes in protein turnover—especially of nuclear proteins—is never assessed.

- o On p. 7, the authors write, “Altogether, our results provide compelling evidence that naïve cells undergo substantial post-transcriptional control through degradation of LIN28, releasing mRNAs from its repressive control.” However, the role of LIN28 is never directly tested. The authors can speculate that LIN28 may play a role, but absent functional manipulation of LIN28 to test this hypothesis, these conclusions remain premature. This point is relevant again in the discussion where the authors write that the metabolic “switch is largely controlled post-transcriptionally by RNA binding proteins. Degradation of LIN28 in both 2i or CDK8/19i, releases hundreds of mRNA from its repressive influence, enabling their translation.” This hypothesis is not directly tested and so the conclusions should be modified.

- o On p. 8, the authors note no change in expression of DNA methyltransferases and demethylases, “which confirms that repression of de novo methylation is not the differential factor that explains hypomethylation in 2i-ESCs in the face of preservation of methylation in CDK8/19i-ESCs (Figure 4E). On the other hand, TET dioxygenases, AICDA and TGA exhibited similar regulation, discarding a differential effect too (Figure 4E).” However, many factors can regulate activity of these enzymes (including SAM, as the authors themselves note!), and so the conclusion that these enzymes are not involved is overstated and should be revised.

o On p. 10, “These results show that 2i and CDK8/19i caused a similar inhibition on the mTORC1 pathway and explain the increased autophagic flux described above in our data.” The role of mTORC1 in autophagy is not tested.

o The revised text covering the ERF data greatly clarifies the results and is appreciated. However, the authors at several points conflate correlation with causation, concluding that the data “confirm[...] the role of ERF as an early downstream effector” of 2i and provide “further evidence of the functional role of ERF as a repressor in maintenance of the naïve state.” Likewise, many other factors beyond ERF could contribute to regulation of genes such as KLF4, ESRRB, etc. Absent genetic data testing the function of ERF, these conclusions cannot be drawn. This becomes a larger issue on p. 13 when upregulation of the same pluripotency proteins are attributed to CDK8/19-specific regulation of super-enhancer containing genes. The authors conclude that the same family of genes is regulated by super enhancers in CDK8/19i treated cells but by ERF nuclear translocation in 2i. Neither of these hypotheses are tested directly and so these conclusions and discussion should be modified.

Other points:

. The authors did not address Reviewer 1’s request that key findings be repeated in the absence of serum. While it is appreciated that the major features of naïve pluripotency can be achieved in the presence of serum, demonstrating that select key changes also occur in the absence of serum is a fair request consistent with other publications that include a serum-free condition for key findings.

. The suggestion that “glycine-derived threonine is the preferred 1C donor through the glycine cleavage system” rather than glucose-derived serine is too premature based on the current data. Also do the authors mean to refer to threonine derived glycine, rather than the inverse?

REVIEWERS' COMMENTS

Reviewer #1 (Remarks to the Author):

With respect to the paper “Dissection of two routes to naïve pluripotency using different kinase inhibitors” by Martinez-Val et al.”, I think the authors significantly improved their manuscript, it has become a very thorough and insightful manuscript.

There are however three final issues that should be addressed before publication:

- IMPORTANTLY: I forgot to mention in my previous review, but then and now the mass spectrometry proteomics data are not available in PRIDE using the reviewer login credentials, they seem to not have been submitted (“No Review Data”). All data (metadata AND processed data) should be submitted to a public database, and immediately made open access upon publication of this manuscript.
- In Fig 3a and Fig 3g (and elsewhere if not provided), the authors should add protein names to their heatmaps.
- I am very surprised that authors indicate that they much appreciate and value my comments, perform subsequent data analysis in the rebuttal to tackle the issue, but do not show these actual analyses/ figures anywhere in the paper (in particular if this happens multiple times). I feel the authors should seriously consider including the figures as shown in the rebuttal (1. Performed on my question “2) I was very surprised to read”; 2. Performed on my question “3) The analysis and representation of the phosphoproteomics is”; 3. Performed on the question of Reviewer 3 point a: “a. The authors point”). These are all highly useful analyses for readers that are likely having similar concerns as the reviewers have.

Reviewer #2 (Remarks to the Author):

The authors have addressed my concerns appropriately, and the new data have strengthened the manuscript. I therefore recommend acceptance for publication.

Reviewer #3 (Remarks to the Author):

In this revised manuscript, the authors add some new data, including measures of LC3 lipidation, ERF localization, and complex I dependency; include additional discussion, and temper the interpretation of the data, resulting in an improved submission. However, as pointed out by Reviewer 1 in the original submission, this paper remains a data resource paper without functional validation of any key findings. Whether any of the implicated pathways contribute to gene/protein regulation in ESCs is not tested. A prerequisite to publication is further modification of the text to ensure that no conclusions are drawn from correlative data. Once such critical modifications, detailed below, are included, this resource will be of interest to pluripotent stem cell enthusiasts.

Overall, the authors have done a good job modulating many of their conclusions to reflect their data. However, there remain many examples where the authors make conclusions based on correlative data. Some instances of drawing functional conclusions from descriptive data persist, and must be modified to reflect the actual data. Examples include:

- o P.5 the addition of LC3B-II levels is a welcome addition to demonstrate potential alterations in autophagy, but steady state levels of LC3 lipidation can reflect either changes in autophagosome synthesis or degradation and thus do not reflect flux (among other examples, see PMID: 25484088). The authors thus should revise their conclusions and avoid any implication that “increased autophagic flux is a common feature of naïve pluripotency.”

- o On p.5, the authors write, “our results reveal the rapid and concerted activation of a proteostasis axis induced under the early effects of 2i, leading to the degradation of nuclear proteins”. However, direct assessment of changes in protein turnover—especially of nuclear proteins—is never assessed.

- o On p. 7, the authors write, “Altogether, our results provide compelling evidence that naïve cells undergo substantial post-transcriptional control through degradation of LIN28, releasing mRNAs from its repressive control.” However, the role of LIN28 is never directly tested. The authors can speculate that LIN28 may play a role, but absent functional manipulation of LIN28 to test this hypothesis, these conclusions remain premature. This point is relevant again in the discussion where the authors write that the metabolic “switch is largely controlled post-transcriptionally by RNA binding proteins. Degradation of LIN28 in both 2i or CDK8/19i, releases hundreds of mRNA from its repressive influence, enabling their translation.” This hypothesis is not directly tested and so the conclusions should be modified.

- o On p. 8, the authors note no change in expression of DNA methyltransferases and demethylases, “which confirms that repression of de novo methylation is not the differential factor that explains hypomethylation in 2i-ESCs in the face of preservation of methylation in CDK8/19i-ESCs (Figure 4E). On the other hand, TET dioxygenases, AICDA and TGA exhibited similar regulation, discarding a differential effect too (Figure 4E).” However, many factors can regulate activity of these enzymes (including SAM, as the authors themselves note!), and so the conclusion that these enzymes are not involved is overstated and should be revised.

- o On p. 10, “These results show that 2i and CDK8/19i caused a similar inhibition on the mTORC1 pathway and explain the increased autophagic flux described above in our data.” The role of mTORC1 in autophagy is not tested.

- o The revised text covering the ERF data greatly clarifies the results and is appreciated. However, the authors at several points conflate correlation with causation, concluding that the data “confirm[...] the role of ERF as an early downstream effector” of 2i and provide “further evidence of the functional role of ERF as a repressor in maintenance of the naïve state.” Likewise, many other factors beyond ERF could contribute to regulation of genes such as KLF4, ESRRB, etc. Absent genetic data testing the function of ERF, these conclusions cannot be drawn. This becomes a larger issue on p. 13 when upregulation of the same pluripotency proteins are attributed to CDK8/19-specific regulation of super-enhancer containing genes. The authors conclude that the same family of genes is regulated by super enhancers in CDK8/19i treated cells but by ERF nuclear translocation in 2i. Neither of these hypotheses are tested directly and so these conclusions and discussion should be

modified.

Other points:

. The authors did not address Reviewer 1's request that key findings be repeated in the absence of serum. While it is appreciated that the major features of naïve pluripotency can be achieved in the presence of serum, demonstrating that select key changes also occur in the absence of serum is a fair request consistent with other publications that include a serum-free condition for key findings.

. The suggestion that "glycine-derived threonine is the preferred 1C donor through the glycine cleavage system" rather than glucose-derived serine is too premature based on the current data. Also do the authors mean to refer to threonine derived glycine, rather than the inverse?

RESPONSE TO REVIEWERS

Please, note all references to changes and modifications to the text and figures are now based on the new updated versions of the manuscript files. Also, all the line numbering below is presented with "Track Changes: Enabled" and "Display for Review: No Markup" in Microsoft Word.

Reviewer #1 (Remarks to the Author):

With respect to the paper "Dissection of two routes to naïve pluripotency using different kinase inhibitors" by Martinez-Val et al.", I think the authors significantly improved their manuscript, it has become a very thorough and insightful manuscript.

We thank this reviewer for the positive comments and feedback.

There are however three final issues that should be addressed before publication:

- IMPORTANTLY: I forgot to mention in my previous review, but then and now the mass spectrometry proteomics data are not available in PRIDE using the reviewer login credentials, they seem to not have been submitted ("No Review Data"). All data (metadata AND processed data) should be submitted to a public database, and immediately made open access upon publication of this manuscript.

We apologize for this mistake. It seems there was an unnoticed issue with the username and password originally received from ProteomeXchange. This has now been solved. The proteomic and phosphoproteomic data can be accessed with the same accession number PXD018694 using the following username (reviewer97575@ebi.ac.uk) and password (Yj9DOXfj). On the other hand, the raw metabolomic data have been deposited in the Metabolights database with the Study ID: MTBLS301. URL for review purposes:

<https://www.ebi.ac.uk/metabolights/reviewer0676eac852439e4fad89e166b014012e>

- In Fig 3a and Fig 3g (and elsewhere if not provided), the authors should add protein names to their heatmaps.

We have now included protein names in Fig 3.

- I am very surprised that authors indicate that they much appreciate and value my comments, perform subsequent data analysis in the rebuttal to tackle the issue, but do not show these actual analyses/ figures anywhere in the paper (in particular if this happens multiple times). I feel the authors should seriously consider including the figures as shown in the rebuttal

1. Performed on my question “2) I was very surprised to read”;
2. Performed on my question “3) The analysis and representation of the phosphoproteomics is”;
3. Performed on the question of Reviewer 3 point a: “a. The authors point”).

These are all highly useful analyses for readers that are likely having similar concerns as the reviewers have.

Indeed, we valued the previous positive criticisms from all reviewers and we believe these suggestions and comments significantly improved our manuscript. In the first revision, we felt to include only the re-analysis of Atlasi et al data (PMID: 32238817) with regard to the presence of post-transcriptional regulation because of its prominent role in our work, and because this issue was raised by both Reviewers 1 and 3. We acknowledge that the other re-analyses are also important and valuable for the potential readers of this work, and we are happy to include these as Supplementary Figures and Supplementary Notes (to comply with the length constraints of the journal and also following the editor’s suggestions). Specifically:

- The reanalysis of our published RNA-seq data in serum-free and serum-containing media is now in Supplementary Note 1.
- The re-analysis of Atlasi et al. data with respect to changes in mRNA/RFP/protein levels is now included as Supplementary Note 2.
- The comparisons of phosphoproteomic and proteomic changes are now included in Supplementary Note 3.
- The re-analysis of Atlasi et al. data with respect to changes in autophagy components is now included as Supplementary Fig 2c.

Reviewer #2 (Remarks to the Author):

The authors have addressed my concerns appropriately, and the new data have strengthened the manuscript. I therefore recommend acceptance for publication.

We thank this reviewer for recommending our manuscript for publication.

Reviewer #3 (Remarks to the Author):

In this revised manuscript, the authors add some new data, including measures of LC3 lipidation, ERF localization, and complex I dependency; include additional discussion, and temper the interpretation of the data, resulting in an improved submission. However, as pointed

out by Reviewer 1 in the original submission, this paper remains a data resource paper without functional validation of any key findings. Whether any of the implicated pathways contribute to gene/protein regulation in ESCs is not tested. A prerequisite to publication is further modification of the text to ensure that no conclusions are drawn from correlative data. Once such critical modifications, detailed below, are included, this resource will be of interest to pluripotent stem cell enthusiasts.

We appreciate that this reviewer recognizes the improvements on the manuscript after the first revision. Our work is conceived as an integrative multi-omics characterization of two chemical approaches that stabilize naive pluripotency. As such, it contains a description of the observed molecular events, which in agreement with this reviewer, we also think it can be a valuable resource for other colleagues working in the field. At the same time, we fully agree with this reviewer that several of our conclusions are based on merely correlative data and, perhaps, we have over-interpreted some results without further experimental demonstration. We have now thoroughly revised the text and moderated several of our conclusions. We have put special emphasis on the six points described below but we have also adjusted some other statements to more accurately reflect our data. All these changes are highlighted in yellow in this revised version. We thank this reviewer for these comments, which we believe introduce the data in a more neutral way.

Overall, the authors have done a good job modulating many of their conclusions to reflect their data. However, there remain many examples where the authors make conclusions based on correlative data. Some instances of drawing functional conclusions from descriptive data persist, and must be modified to reflect the actual data. Examples include:

- o P.5 the addition of LC3B-II levels is a welcome addition to demonstrate potential alterations in autophagy, but steady state levels of LC3 lipidation can reflect either changes in autophagosome synthesis or degradation and thus do not reflect flux (among other examples, see PMID: 25484088). The authors thus should revise their conclusions and avoid any implication that “increased autophagic flux is a common feature of naïve pluripotency.”

We thank this reviewer for pointing out this mistake, and for suggesting this reference, which we found very valuable. Indeed, the accurate and quantitative assessment of autophagic flux is complex and requires measuring the autophagosome accumulation rate after fusion inhibition. Therefore, the increased signal in LC3-II detected in our western blot could indicate either enhanced synthesis or decreased degradation of autophagosomes. In this revised version, we have clarified this issue (Pages 4-5, lines 124-128). The header of this section has also been adjusted to present the results in a more impartial way (Page 4, line 106).

- o On p.5, the authors write, “our results reveal the rapid and concerted activation of a proteostasis axis induced under the early effects of 2i, leading to the degradation of nuclear proteins”. However, direct assessment of changes in protein turnover—especially of nuclear proteins—is never assessed.

We acknowledge that the link between the up-regulation of proteostasis components and degradation of nuclear proteins is not demonstrated. Consequently, we have now re-phrased

this section, presenting these observations as a hypothesis that will certainly require experimental validation (Page 5, lines 141-146)

o On p. 7, the authors write, “Altogether, our results provide compelling evidence that naïve cells undergo substantial post-transcriptional control through degradation of LIN28, releasing mRNAs from its repressive control.” However, the role of LIN28 is never directly tested. The authors can speculate that LIN28 may play a role, but absent functional manipulation of LIN28 to test this hypothesis, these conclusions remain premature. This point is relevant again in the discussion where the authors write that the metabolic “switch is largely controlled post-transcriptionally by RNA binding proteins. Degradation of LIN28 in both 2i or CDK8/19i, releases hundreds of mRNA from its repressive influence, enabling their translation.” This hypothesis is not directly tested and so the conclusions should be modified.

We agree that the role of LIN28 in the repression of post-transcriptionally controlled genes is not demonstrated in our work and is only inferred on the basis of correlative data. Previously, LIN28 has been shown to repress mRNA translation in mESCs for mitochondria (PMID: 27320042) and endoplasmic reticulum (PMID: 23102813). Importantly, some of the LIN28 targets described in such publications were found in our data to exhibit increased protein levels, compared to their cognate mRNAs (e.g. Lamp1, Cdh1, Uqcrcf1). Our hypothesis therefore seems to agree with such studies and present other potential targets that might be regulated in a similar manner. Because this is indeed never proven in our data, we have now moderated our conclusions in section header (Page 5, Line 147), results (Page 6, lines 195-196) and discussion (Page 13, lines 431-432 and lines 439-441).

o On p. 8, the authors note no change in expression of DNA methyltransferases and demethylases, “which confirms that repression of de novo methylation is not the differential factor that explains hypomethylation in 2i-ESCs in the face of preservation of methylation in CDK8/19i-ESCs (Figure 4E). On the other hand, TET dioxygenases, AICDA and TGA exhibited similar regulation, discarding a differential effect too (Figure 4E).” However, many factors can regulate activity of these enzymes (including SAM, as the authors themselves note!), and so the conclusion that these enzymes are not involved is overstated and should be revised.

The referee is again right regarding this point. Our conclusions regarding the factors that determine the hypomethylated DNA of 2i are only based on the expression changes of these proteins and many other factors could also influence their activities. In fact, in the work from von Meyenn et al. (PMID: 27237052) the authors found out that TET dioxygenases seem to have a role in the active de-methylation typical in the serum-2i conversion, although the authors themselves remark that this activity is neither sufficient nor necessary for this process to occur. Given the fact that we found similar changes in maintenance methylation proteins, our data agree with von Meyenn results, without neglecting the potential contribution of de novo methylation and active demethylation. We have reformulated this conclusion to make this point clearer (Page 8, lines 247-249). Accordingly, we have also tempered the discussion of these results (Page 14, line 462-465).

o On p. 10, “These results show that 2i and CDK8/19i caused a similar inhibition on the mTORC1 pathway and explain the increased autophagic flux described above in our data.” The role of mTORC1 in autophagy is not tested.

We agree that our data do not demonstrate that the dephosphorylation of mTOR observed in our data leads to increased autophagy (our claim of increased autophagic flux has also been corrected in the above comment). We have now removed this sentence from the results (Page 10, line 309) and adjusted the discussion section accordingly (Page 14, lines 446-447).

o The revised text covering the ERF data greatly clarifies the results and is appreciated. However, the authors at several points conflate correlation with causation, concluding that the data “confirm[...] the role of ERF as an early downstream effector” of 2i and provide “further evidence of the functional role of ERF as a repressor in maintenance of the naïve state.” Likewise, many other factors beyond ERF could contribute to regulation of genes such as KLF4, ESRRB, etc. Absent genetic data testing the function of ERF, these conclusions cannot be drawn. This becomes a larger issue on p. 13 when upregulation of the same pluripotency proteins are attributed to CDK8/19-specific regulation of super-enhancer containing genes. The authors conclude that the same family of genes is regulated by super enhancers in CDK8/19i treated cells but by ERF nuclear translocation in 2i. Neither of these hypotheses are tested directly and so these conclusions and discussion should be modified.

We appreciate that our previous modifications have clarified the analyses of ERF data. Yet, we agree with this reviewer that these data are still merely correlative and these transcriptional relationships cannot be inferred without further experimental validation. We have toned down our interpretation of these results (Page 11, lines 361-362 and lines 368-372). We have also re-phrased the Header section (Page 11, line 350). We have also adjusted the text concerning the transcriptional changes observed in CDK8/19i (Page 13, line 406 and lines 412-415). Lastly, we have also clarified in the Discussion section that this is a proposed model derived from the integration of our data sets (Page 13, lines 419-424).

Other points:

. The authors did not address Reviewer 1’s request that key findings be repeated in the absence of serum. While it is appreciated that the major features of naïve pluripotency can be achieved in the presence of serum, demonstrating that select key changes also occur in the absence of serum is a fair request consistent with other publications that include a serum-free condition for key findings.

Because we aimed to study the direct effects of MEKi/GSK3i vs CDK8/19i on naive pluripotency, we chose to keep serum in our cultures to avoid the potential confounding effects of removing serum in our comparisons. In our preceding publication, we showed that several of the features in 2i/LIF (serum-free, KSR) were recapitulated in CDK8/19i (serum-free, KSR) (PMID: 32989249). In addition, in the previous revision of our current manuscript, we re-analyzed RNA-seq data from 2i and CDK8/19i cells in serum and serum-free conditions and showed that both settings induce similar transcriptional changes. Further, we found evidence of post-transcriptional regulation in mitochondrial proteins in the data from Atlasi et al (which compared serum/LIF vs 2i/LIF) as well as in the up-regulation of autophagy proteins. Therefore, we feel that these analyses demonstrate that some of the changes described in our manuscript

are reproduced in serum-free conditions. Following Reviewer 1 suggestions, we have now included these re-analyses as Supplementary Note 1.

. The suggestion that “glycine-derived threonine is the preferred 1C donor through the glycine cleavage system” rather than glucose-derived serine is too premature based on the current data. Also do the authors mean to refer to threonine derived glycine, rather than the inverse? This reviewer is right. There was a mistake in this sentence because, indeed, we referred to “threonine-derived glycine”. Nevertheless, we agree with the reviewer that this statement cannot be supported only based on the expression changes of the enzymes involved in this pathway. In this revised version, we have removed this conclusion from the discussion. We have also adjusted the text of this section to present these data in a more neutral way (Page 14, lines 452-460).